# Modelling the terrestrial nitrogen and phosphorus cycle in the UVic ESCM

Makcim De Sisto [1,2], Andrew H. MacDougall [1], Nadine Mengis [3], and Sophia Antoniello [1]

[1]St. Francis Xavier University, Antigonish, NS, Canada
[2]Faculty of Engineering and Applied Science, Memorial University of Newfoundland, NL, Canada
[3]GEOMAR Helmholtz Centre for Ocean Research Kiel, Kiel, Germany

**Correspondence:** Makcim De Sisto (mdesisto@stfx.ca)

**Abstract.** Nitrogen (N) and Phosphorus (P) biogeochemical dynamics are crucial for the regulation of the terrestrial carbon cycle. In Earth System Models (ESMs) the implementation of nutrient limitations has been shown to improve the carbon cycle feedback representation and hence, improve the fidelity of the response of land to simulated atmospheric $CO_2$ rise. Here we aimed to implement a terrestrial N and P cycle in an Earth system model of intermediate complexity to improve projections of the future $CO_2$ fertilization feedbacks. The N cycle is an improved version of the Wania et al. (2012) N module, with enforcement of N mass conservation and the merger with a deep land-surface and wetland module that allows for the estimation of $N_2O$ and NO fluxes. The N cycle module estimates fluxes from three organic (litter, soil organic matter and vegetation) and two inorganic ($NH_4^+$+and $NO_3^-$ ) pools, accounts for inputs from biological N fixation and N deposition. The P cycle module contains the same organic pools with one inorganic P pool, it estimates influx of P from rock weathering and losses from leaching and occlusion. Two historical simulations are carried for the different nutrient limitation setups of the model: Carbon and Nitrogen (CN) and Carbon, Nitrogen and Phosphorus (CNP), with a baseline carbon only simulation. The improved N cycle module now conserves mass and the added fluxes (NO and $N_2O$), along with the N and P pools are within the range of other studies and literature. For the years 2001-2015 the nutrient limitation resulted in a reduction of GPP from the Carbon-only value of 143 PgC yr$^{-1}$ to 130 PgC yr$^{-1}$ in the CN version and 127 PgC yr$^{-1}$ in the CNP version. This implies that the model efficiently represents a nutrient limitation over the $CO_2$ fertilization effect. CNP simulation resulted in a reduction of 11% of the mean GPP and a reduction of 23% of the vegetation biomass compared to baseline C simulation. These results are in better agreement with observations, particularly in tropical regions where P limitation is known to be important. In summary, the implementation of the N and P cycle have successfully enforced a nutrient limitation in the terrestrial system, which now have reduced the primary productivity and the capacity of land to uptake atmospheric carbon better matching observations.

## 1 Introduction

Terrestrial biogeochemical cycles are sensitive to changes in atmospheric $CO_2$ concentrations and climate. Their global evolution will determine the capacity of vegetation and soils to store anthropogenic carbon (Goll et al. , 2012). In terrestrial ecosystems carbon cycle feedbacks are constrained in part by the availability of nutrients (Fisher et al. , 2012; Zaehle et al. , 2014; Wieder et al. , 2015; Du et al. , 2020). Among nutrients Nitrogen (N) and Phosphorus (P) are considered to be the most

critical for limiting the primary productivity (Filipelli , 2002; Fowler et al. , 2013). Both are fundamental functional needs for plant biochemistry and their requirement is common in all vegetation taxa (Filipelli , 2002; Vitousek et al. , 2010; Du et al. , 2020). Regionally, the availability of nutrients can impair the photosynthetic efficiency of terrestrial vegetation and consequently their response to increasing atmospheric $CO_2$. Hence, in Earth System Models (ESMs) the representation of nutrient limitations is an imperative to improve the accuracy of carbon feedback projections and estimation of carbon budgets.

The simulations from first generation ESMs with carbon only schemes, have very likely overestimated the response of the terrestrial ecosystem to the increase of atmospheric $CO_2$ concentrations (Hungate et al. , 2003; Thorton et al. , 2007), showing a high terrestrial carbon uptake response which would require an unrealistic large nutrient supply. The addition of a N cycle to the land system in ESMs has shown an overall reduction in the effect of $CO_2$ fertilization especially in high latitudes, with a weaker response in low latitudes which are typically P limited in natural systems (Wang et al. , 2007, 2010; Goll et al. , 2017;

Du et al. , 2020; Wang et al. , 2020).

The global distribution of N and P is dependent on the biogeochemical characteristics of each nutrient. N inputs are mainly from Biological Nitrogen Fixation (BNF) and atmospheric deposition with little addition from rock weathering (Du et al. , 2020). There are two types of N deposition from the atmosphere: wet (precipitation) and dry (particles). Among the two, wet deposition represents most of the atmospheric N input (Fowler et al. , 2013; Dynarski et al. , 2019). In contrast, the main

input of P comes from rock weathering (mainly apatite) with lesser inputs from atmospheric deposition as dust particles. These characteristic are among the reasons of a global spatial pattern where young soils are usually N limited and old soil are P limited (Filipelli , 2002; Fowler et al. , 2013; Du et al. , 2020). N accumulates rapidly from BNF where N fixers are abundant and slowly where atmospheric deposition is dominant. Thereby, old soils have a larger accumulation of N especially in regions where N fixers are abundant. On other hand, P input is limited by the parent material and the bioavailability is further constrain

by the retention of recalcitrant P in soils. Walker and Syers  (1976) even suggested that P storage has a fix total that cannot be rapidly replenished as parent material is limited.

These notions led to the common conceptualization that high latitudes are N limited while tropical regions are P limited. While this generalization is correct in most observational studies, the complex pattern of limitation is more intricate, and P limitation could be more common than is commonly inferred. Du et al.  (2020) found that globally 43% of the terrestrial system

are relatively limited by P while only 18% limited by N with the rest being co-limited by both. Biochemically, the availability of N and P can directly limit on another. The addition of P has been shown to be positive for the N fixation, leading to the replenishment of N is ecosystems (Eisele et al. , 1989). N supply on the other hand regulates the production of the enzyme phosphatase that cleaves ester-P bonds in soil organic matter (McGill and Cole , 1981; Olander and Vitousek , 2000; Wang et al. , 2007).

Biodiversity plays a crucial role in biogeochemical cycles. The fluxes and availability of N and P in soils depends on the interactions between soil mineral matrix, plants and microbes (Cotrufo et al. , 2013). For example, N input from atmospheric $N_2$ fixation is mediated by a specialized group of microorganisms. Furthermore, the recycling on N from plants-soil-microbes determines the availability of N for plant uptake. Overall, the land biota dynamics impacts the productivity, ecosystem resiliense and stability (Yang et al. , 2018). High diversity has been linked to enhanced vegetation productivity (Wagg et al. , 2014). The

diversity in terrestrial ecosystem is determined by biological, environmental and physico-chemical processes. Anthropogenic activities can influence soil diversity, impacting the availability and cycling of N and P (Chen et al. , 2019). For example, N and P fertilization, have been shown to affect soil microbial biomass and composition (Ryan et al. , 2009). Plant diversity, is linked to soil health and functioning, and is core for the N and P cycles. Vegetation species variable adaption to nutrient concentrations also plays a role in the availability of nutrients in soils and the biogeography of terrestrial vegetation. Overall, biodiversity constitute an environmental resilience factor to abrupt changes (Van Oijen et al. , 2020). However, implementing such dynamics remains far beyond the capabilities for the present generation Earth systems models. Several studies have found that in some ecosystems lack of N in soil usually leads to dominance of woody symbiotic N fixers (e.g. Menge et al. , 2012). The availability of P is also impacted by the geochemical interactions in terrestrial soils, Vitousek et al. (2010) defined six mechanisms by which P is driven to limitation: loss by leaching, soil barriers that physically prevents access to roots, slow release of mineral P forms, P parent material, sequestration of P in soils and pools in the ecosystem and finally, anthropogenic input of nutrients.

Despite its importance P terrestrial limitation has been rare in Earth system modelling. The effect of P in tropical forest may be the key to better represent the vegetation biomass and the response to $CO_2$ fertilization. The lack of P observational data is in partly responsible for the difficulty of simulating P limitation in Earth system models (Spafford and MacDougall , 2021). However, several studies have attempted to provide reliable global P datasets (Yang et al. , 2013; Hartmann et al. , 2014; He et al. , 2021) that could be use to develop more accurate models. Furthermore, many studies have shown that the inclusion of P into ESM structures is possible and that it improves the representation of vegetation biomass in tropical regions (Wang et al. , 2007, 2010; Goll et al. , 2012, 2017; Fleischer et al. , 2019; Thum et al. , 2019; Yang et al. , 2019; Wang et al. , 2020; Nakhavali et al. , 2021). The addition of nutrient limitation has been observed to mainly effect the capacity of vegetation to uptake carbon (Wang et al. , 2010; Goll et al. , 2017; Wang et al. , 2020). Therefore, the accumulation of carbon in the atmosphere is enhanced, leading to increases of temperature in simulations. This temperature changes are likely to have some impact to variables sensitive to atmospheric temperature changes. Furthermore, the decrease of vegetation biomass affects variables affected by the distribution and composition of plant functional types, as changes in terrestrial albedo.

Intermediate complexity Earth system models, have a lower spatial representation, and model structures that have been intentionally simplified in one or more ways. This simplification allows for long-term simulations that are typically not feasible in higher complexity models. This class of model is not suitable for studying processes at small spatial scales. Hence, they are used in research questions that require large spatial and temporal scales (Weber , 2010). Current generation Earth system models are or have already developed nutrient limitation to their model structure (e.g., Community Land Model (Lawrence et al. , 2019), Joint UK Land Environment Simulator (Clark et al. , 2011), Community Atmosphere–Biosphere Land Exchange model (Haverd et al. , 2018), Australian Community Climate and Earth System Simulator (Ziehn et al. , 2020)). While CN models are more common CNP models remains to be rarer. However, P cycles have been suggested to be included into Earth system model for its importance in tropical regions (Wang et al. , 2010; Goll et al. , 2012). The first attempt to include nutrient limitation in the University of Victoria Earth system and climate model (UVic ESCM) was done by Wania et al. (2012) but was not included in the current publically available version of the model due to the need of further improvement. We aim to

describe a terrestrial N and P cycle adapted, developed and implemented for the UVic ESCM version 2.10. The main dynamics captured in this study are in the terrestrial system, especially vegetation. Furthermore, we intent to improve the current state of the previous N cycle implement in the UVic ESCM, develop a new P cycle and couple carbon N and P, in order to improve the carbon cycle feedbacks projections.

## 2 Methodology

### 2.1 Model description

The UVic ESCM is a climate model of intermediate complexity (ver. 2.10, Weaver et al. (2001); Mengis et al. (2020)), it contains a simplified moisture-energy balance atmosphere coupled with a three-dimensional ocean general circulation (Pacanowski , 1995) and a thermodynamic sea-ice model (Bitz et al. , 2001). The model has a common horizontal resolution of 3.6° longitude and 1.8° latitude and the oceanic module has a vertical resolution of 19 levels with a varying vertical thickness (50 m near the surface to 500 m in the deep ocean).

In version 2.10, the soil is represented by 14 subsurface layers with thickness exponentially increasing with depth with a surface layer of 0.1 m, a bottom layer of 104.4 m and a total layer of 250 m. Only the first 8 layers have active hydrological processes (top 10 m), below that lays bedrock with thermal characteristics of granitic rocks. The soil carbon cycle is active in the top 6 layers up to a depth of 3.35 m (Avis , 2012; MacDougall et al. , 2012) the soil respiration is a function of temperature and moisture (Meissner et al. , 2003). The terrestrial vegetation is simulated by a top-down representation of interactive foliage and flora including dynamics (TRIFFID) representing vegetation interaction between 5 functional plant types: broadleaf trees, needleleaf trees, shrubs, C3 grasses, and C4 grasses that compete for space in the grid following the Lotka-Volterra equations (Cox , 2001). Net carbon fluxes estimated in the model updates the total areal coverage, leaf area inxes and canopy height for each PFT. For each PFT the carbon fluxes are derived from a photosyntesis-stomatal conductance model (Cox et al. , 1998). The carbon uptake though photosynthesis is allocated into growth and respiration and the vegetation carbon is transferred to the soil via litter fall and allocated in the soil as a decreasing function of depth (proportional to root distribution) and expect for the top layer is only added to soil layers with temperature above 1°C.

Furthermore, permafrost carbon is prognostically generated within the model using a diffusion-based scheme meant to approximate the process of cryoturbation (MacDougall and Knutti , 2016). The sediment processes are modelled using an oxic-only calcium carbonate scheme (Archer , 1996). Terrestrial weathering is diagnosed from the spin-up net sediment flux and stays fixes at the preindustrial equilibrium value (Meissner et al. , 2012). Mengis et al. (2020) merged previous version of the UVic ESCM and evaluated its performance representing carbon and heat fluxes, water cycle and ocean tracers. A full description of the model can be found in Mengis et al. (2020).

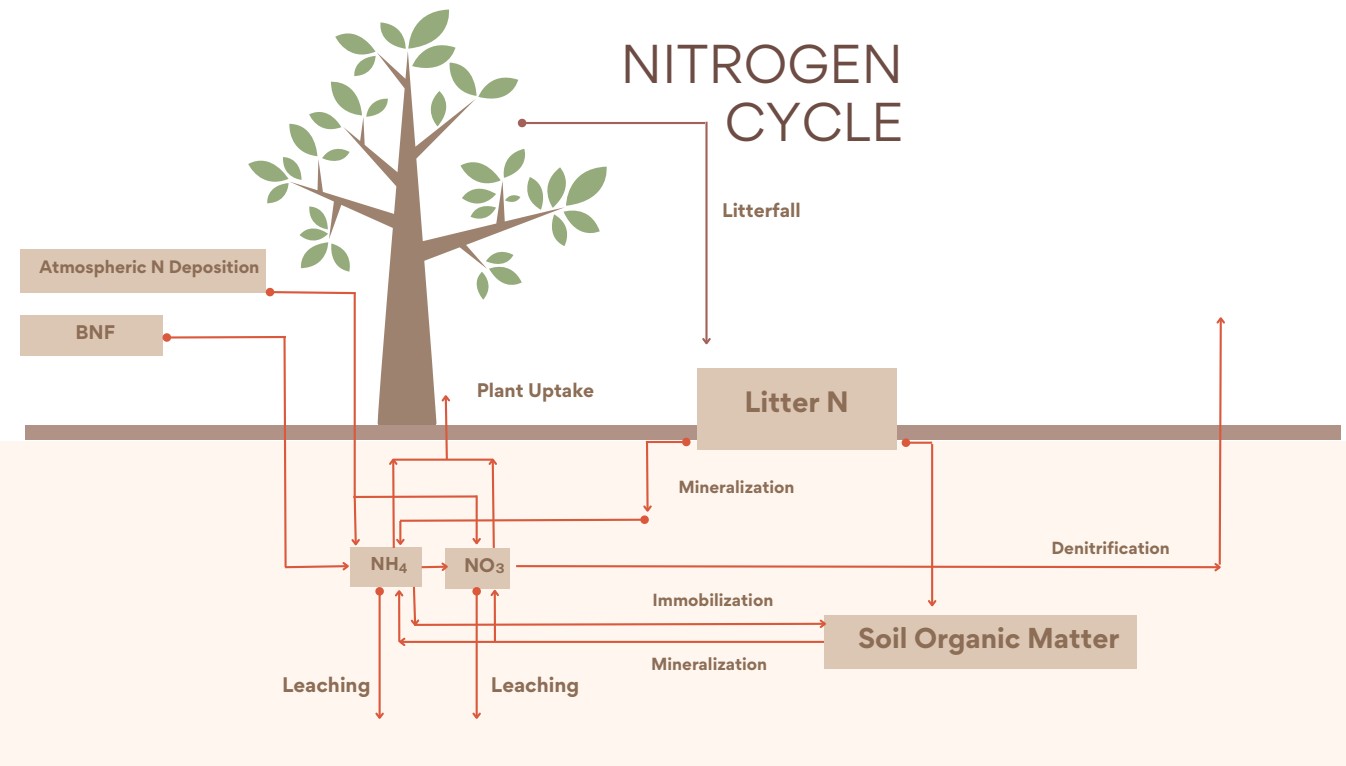

**Figure 1.** Diagram representing the UVic ESCM nitrogen cycle.

## 2.2 Nitrogen cycle

### 2.2.1 Nitrogen uptake

The new N cycle module was adapted from Wania et al. (2012). The module contains three organic (litter, soil organic matter and vegetation) and two inorganic (NH4$^+$, NO3$^-$) N pools. The base structure is based on Gerber et al. (2010) with further modifications to fit the UVic ESCM scheme. NH4$^+$ is produced both from BNF and mineralization of organic N, it can be taken up by plants (vegetation), leached, or transformed into NO3$^-$ via nitrification. NO3$^-$ is produced through nitrification, can be taken up by plants, leached or denitrified into NO, N$_2$O or N$_2$. The inorganic N is distributed between leaf, root and wood, with wood having a fixed stoichiometry ratio and variables ratios for the leaf and root pools. Organic N leaves the living pools via litter-fall into the litter pool which is either mineralized or transferred to the organic soil pool, part of this N can be mineralized into the inorganic N pools. At the same time N can flow from the inorganic to the soil organic pool via immobilization. The CN ratios in leafs are determined by Eq. (1):

$$CN_{leaf} = \frac{C_{leaf}}{N_{leaf}}, \tag{1}$$

where $C_{leaf}$ is the carbon content in leafs and $N_{leaf}$ is the N content in leafs. $CN_{leaf}$ is one of the most important nutrient limitators in the model. It controls the maximum carboxilation rate of RuBISCO. Furthermore, it control vegetation biomass. If leaf C:N ratio is higher than $CN_{leafmax}$ (the maximum CN ratio parameter) terrestrial vegetation biomass is reduced.

The new version of the N cycle has been merged with a deep land-surface (MacDougall and Knutti , 2016) and a new wetland
module (Nzotungicimpaye et al. , 2021). Both inorganic N pools are transferred between soil layers following ground-water flow. Given this flow the distribution of N in layers was taken into account in the uptake calculations in Eq. (2) and (3), a root fraction was added (4) fixing the amount of root biomass per PFTs per layer depth. The equations governing N uptake are:

$$NH_4^{UP} = \sum_{PFT} ( \frac{V_{maxn} C_{root} [NH_4(av)] Froot}{K_{n,1/2} + [Nmin(av)]} + [NH_4(av)] * Qt), \quad (2)$$

$$NO_3^{UP} = \sum_{PFT} ( \frac{V_{maxn} C_{root} [NO_3(av)] Froot}{K_{n,1/2} + [Nmin(av)]} + [NO_3(av)] * Qt), \quad (3)$$

where $NH_4^{UP}$ and $NO_3^{UP}$ represent the N uptake, the left term is the active uptake while the right term is the passive uptake (see table 1), the latter is the transport of N via the transpiration water stream. $V{maxn}$ is the maximum uptake rate for N, $C{root}$ is the root carbon biomass, $[NH_4(av)]$, $[NO_3(av)]$ and $[Nmin(av)]$ are the $NH_4$, $NO_3$ and mineral N concentrations, $K_{n,1/2}$ is the half saturation constant for N and $Qt$ is the transpiration rate. $av$ represents the available portion of $NH_4$ and $NO_3$
in soil. This fraction is calculated as the total concentration of $NH_4$ and $NO_3$ divided by sorption factors (10 and 1 respectively) following Wania et al. (2012). The equation for root fraction is:

$$Froot,_{PFT} = \frac{e^{z_{top,n}/d_{r,PFT}} - e^{z_{bot,n}/d_{r,PFT}}}{1 - e^{D/d_{r,PFT}}} , \quad (4)$$

where $Z_{top}$ and $Z_{bot}$ represents the top layer and bottom layer depth respectively, while D and $d_r$ are the depth of the soil layer and the root depth. The depth of soil layer represents the depth of each specific soil layer. Root depth is a PFT based
parameter that represents the depth of the roots. Given the multiple soil layer set up, the root fraction modifies the value of root carbon, creating a more realistic representation of the uptake root depth reach for each PFT given the multiple soil layer set up.

### 2.2.2 Denitrification

The N cycle was merged with a wetland module that allowed the estimation of anoxic fractions for each soil layer, based on Gedney and Cox (2003). The anoxic fraction is taken to be the saturated fraction of the soil layer that is shielded from $O_2$
by the saturated soil layer above. The Anoxia representation led to denitrification to be added to the N model, accounting the largest exit pathway for N in the terrestrial biosphere. The anaerobic respiration is estimated from eq. (5):

$$R_{an} = K_{rNO_3} f_t f_m C_s A_f \frac{[NO_3(av)]}{[NO_3(av)] + K_n}, \quad (5)$$

where $R_{an}$ is the anaerobic respiration, $K_{rNO3}$ is the ideal respiration rate via $NO_3$ reduction, $f_t$ and $f_m$ are temperature and moisture functions, $C_s$ is the concentration of organic carbon, $Af$ is the anaerobic fraction, $K_n$ is the half-saturation of N-oxides (Li et al. , 2000). The temperature and moisture soil functions are taken directly from Cox (2001), and are represented by the following equations:

$$f_t = q_{10}^{0.1(t_s-25)}, \tag{6}$$

$$f_m = \begin{cases} 1 - 0.8(S - S_0) & \text{for S} > S_0, \\ 0.2 + 0.5(\frac{S-S_W}{S_0-S_W}) & \text{for } S_W < \text{S} \leq S_0, \\ 0.2 & \text{for S} \leq S_W, \end{cases} \tag{7}$$

where in $f_t$, $q_{10} = 2$ and $t_s$ is the soil temperature in °C. In $f_m$, S is the soil moisture, $S_W$ is the wilting point of soil moisture, $S_0$ is the optimum soil moisture. Fluxes of $N_2O$ and NO to the atmosphere are computed based on the 'leaky-pipe' conceptualization of soil-nitrogen processes (Firestone and Davidson , 1989). In the leaky pipe conceptual model $N_2O$ and NO leak out of reactions of one species of nitrogen into another, namely nitrification ($NH_4$ to $NO_3$) and denitrification ($NO_3$ to $N_2$). The size of the holes is determined by the soil processes. For implementation in the UVic ESCM the size of the holes is fixed but the partitioning ratio between NO and $N_2O$ changes based on water filled pore space of the soil layer. The ratio is parameterized based on an empirical relationship derived by Davidson et al. (2000):

$$\frac{N_2O}{NO} = 10^{2.6S_U-1.66}, \tag{8}$$

where $S_U$ is the water filled pore space. Thus, the model produces a total flux of both NO and $N_2O$ for nitrification and denitrification, which is partitioned between the two species based on the above relationship. The NO flux is added to the atmosphere and redeposited as part of the N deposition flux. The $N_2O$ flux is added to the $N_2O$ pool in the atmosphere which has a characteristic half-life of 90.78 years (Myhre et al. , 2013). Decayed $N_2O$ is assumed to become part of the atmospheric $N_2$ pool.

### 2.2.3 Mass balance N cycle

In Wania et al. (2012) N cycle module, under N limitation ($CN_{leaf} > CN_{leafmax}$) the N available was increased artificially by reducing the leaching by up to 100% and if necessary the immobilization by 50%. These mechanics created an unrealistic increase of N in soils and thereby, defying the mass balance conservation of the module.

Here, the vegetation can no longer uptake extra N from leaching or immobilization under nutrient limitation. Instead, under nutrient limitation wood and root carbon mass is transferred as litter (emulating a dying vegetation) until the correct ratio is met. Section 2.4 presents a detailed explanation of nutrient limitation for N and P.

**Table 1.** Updated nitrogen cycle module pools, rates and variables.

| Variables | Units | Type | Descriptions |
|---|---|---|---|
| $NH_4^{UP}$ | kg N m$^{-2}$ yr$^{-1}$ | Rate | $NH_4$ vegetation uptake |
| $NO_3^{UP}$ | kg N m$^{-2}$ yr$^{-1}$ | Rate | $NO_3$ vegetation uptake |
| Croot | Kg C m$^{-2}$ | Pool | Root carbon |
| $[NH_4(av)]$ | kg N m$^{-3}$ | Pool | Available $NH_4$ concentration |
| $[NO_3(av)]$ | kg N m$^{-3}$ | Pool | Available $NO_3$ concentration |
| Froot | - | Variable | Root fraction |
| $[Nmin(av)]$ | kg N m$^{-3}$ | Pool | Available mineral N concentration |
| $R_{an}$ | kg C m$^{-3}$ s$^{-1}$ | Rate | Anaerobic respiration rate |
| $C_s$ | kg C m$^{-3}$ | Pool | Density of soil carbon in each layer |
| $A_f$ | - | Variable | Anaerobic saturation fraction |
| $N_2O$ | kg N m$^{-2}$ yr$^{-1}$ | Rate | Nitrous oxide flux |
| NO | kg N m$^{-2}$yr$^{-1}$ | Rate | Nitric oxide flux |

**Table 2.** Updated nitrogen cycle parameters. See appendix A.1 for values that vary for each PFT.

| Variables | Units | Value | Description | Source |
|---|---|---|---|---|
| $K_{n,1/2}$ | kg N m$^{-3}$ | 0.003 | Half saturation constant for N uptake | Gerber et al. (2010) |
| $Vmaxn$ | kg N (kg root C$^{-1}$ )yr$^{-1}$ | Varies with PFTs | Maximun uptake rate for N | Wania et al. (2012) |
| $D$ | m | Varies with soil layer | Soil layer depth | MacDougall and Knutti (2016) |
| $Qt$ | m yr$^{-1}$ | Varies with PFTs | Transpiration rate | Wania et al. (2012) |
| $z_{top,n}$ | m | Varies with soil layer | Top layer soil depth | Avis (2012) |
| $z_{bot,n}$ | m | Varies with soil layer | Bottom soil layer depth | Avis (2012) |
| d$_r$ | m | Varies with PFTs | Root depth | Avis (2012) |
| K$_{rNO_3}$ | 10$^{-9}$ s$^{-1}$ | 5 | Soil respiration rate for Nitrate respiration | |
| K$_n$ | kg N m$_{-3}$ | 0.083 | Half saturation constant for N-oxides | Li et al. (2000) |
| CN$_{leafmax}$ | kg C (kg N)$^{-1}$ | Varies with PFTs | Maximum CN ratio | Wania et al. (2012) |

## 2.3 Phosphorus cycle

The P cycle is based on Wang et al. (2007, 2010) and Goll et al. (2017) with some equations where modified from Wania et al. (2012) to have a better consistency with N estimations in the new soil layer model. The module contains four inorganic (labile, sorbed, strongly sorbed and occluded) and three organic P pools: Vegetation (leaf, root and wood), litter and soil organic P.

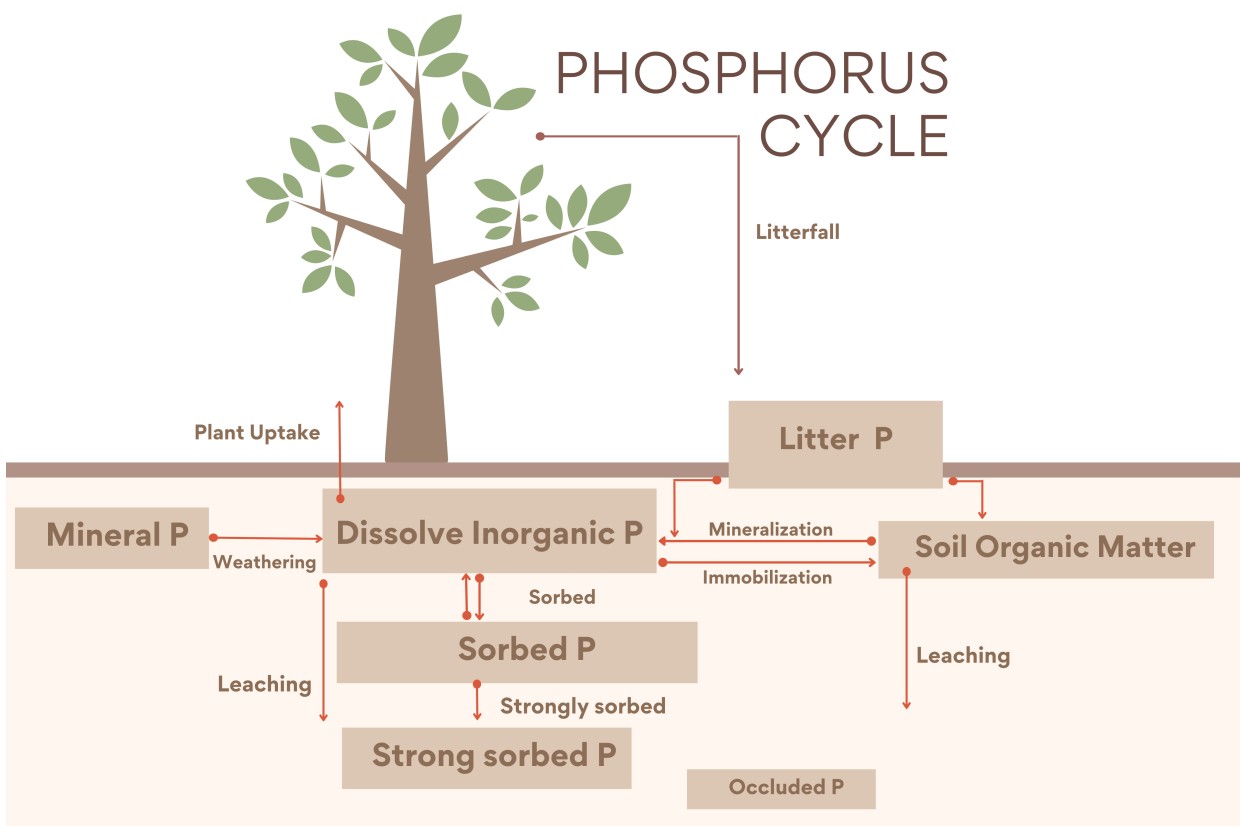

**Figure 2.** Diagram representing the UVic ESCM CNP P cycle. Weathering from mineral P is the only input into the soils. There are 4 inorganic pools (Dissolved inorganic, adsorbed , stronged sorbed and occluded P) and 3 organic pools (vegetation (root, wood and leaf), litter and soil organic matter). As in Wang et al. (2010) the flux from strongly sorbed P to the occluded pool is not represented here, instead it is assumed to be a fraction of total soil P.

### 2.3.1 Input

The P module estimates weathering input following Wang et al. (2010) and is driven by a fixed estimate (Table 3) of P release assigned by soil order divided in 12 classes from U.S. department of agriculture (USDA) soil order map.

Additionally, an extra input structure was tested in the model but was not used for the P results in this study. It was implemented to compare the benefits of a static and a dynamic weathering scheme into the P pool. In this method weathering depends on runoff following Hartmann et al. (2014) using the lithological world map with 16 different classes generated by Hartmann and Moosdorf (2012). Eq. 9 shows the estimation of the chemical weathering rate:

$$F_{CW} = b_i q, \tag{9}$$

**Table 3.** Constants for P input from Wang et al. 2010. The values change depending on the weathering state of the soil type. Highly weathered soils have lower values.

| Soil order | Value (gP m$^{-2}$ yr$^{-1}$) |
|---|---|
| Entisol | 0.05 |
| Inceptisol,Gellisol,Histosol | 0.05 |
| Vertisol | 0.01 |
| Aridisol,Andisol | 0.01 |
| Mollisol | 0.01 |
| Alfisoil,Spodosol | 0.01 |
| Ultisoil | 0.005 |
| Oxisoil | 0.003 |

$$b_i = b_{carbonate} + b_{silicate}, \tag{10}$$

where $F_{CW}$ (t km$^{-2}$ yr$^{-1}$) is the chemical weathering rate, q is the runoff (mm yr$^{-1}$) and $b_i$ is the factor for each lithological class i; shielding correction functions were not applied. The chemical weathering is defined as the total fluvial export of Ca + Mg + K + SiO$_2$ and carbonate derived CO$_3$, b$_{carbonate}$ and b$_{silicate}$ are chemical weathering parameters associated to carbonate and silicate rocks respectively found in Hartmann et al. (2014). Here we only apply Wang et al. (2010) approach as we found it to be more controllable and an advantage to the planned coupling of P flux from land into the ocean. Hartmann et al. (2014) requires the estimation of runoff by the model structure. Hence, while representing a dynamical P release it needs to be carefully assessed so that no extreme overestimation or underestimation are represented regionally. Wang et al. (2010) approach provides constant input without variability which in this particular case is favorable.

## 2.3.2 Inorganic soil phosphorus

Inorganic P (Psoil) in soil follows the dynamics described in (Goll et al. , 2017) in eq. (11), where each time step a fix fraction (k$_s$) of P is adsorbed and the rest is dissolved (1-k$_s$). This fraction is based on Hedley fractionation method (Hedley and Stewart , 1982) which is dependant on soil orders, the dataset has been commonly used to assess the different P forms in soil. The adsorbed P is regulated by k$_s$ in eq. (12) as determined by the soil order in Hedley dataset:

$$\frac{dPsoil}{dt} = (1 - K_s)(P_{wea} + P_{litmin} + P_{orgmin} - P_{leach} - P_{up} - \tau_{sorb}P_{sorb} - P_{imm}), \tag{11}$$

$$\frac{dPsorb}{dt} = K_s \frac{dPsoil}{dt}, \tag{12}$$

where $P_{wea}$ is the P released by rock weathering, $P_{litmin}$ is the P mineralized from the P litter pool, $P_{orgmin}$ is the P mineralized from the soil organic P, $P_{leach}$ is the leached inorganic P, $P_{up}$ is the P uptake by plants, $P_{sorb}$ is the amount of P sorbed, $\tau_{sorb}$ is the rate of strong sorption and $P_{imm}$ is the P immobilized from the inorganic P pool. The estimation of P$soil$ based on Goll et al. (2017), is originally taken from Goll et al. (2012). Here, the sum $P_{sorb}$ and $P_{soil}$ constitute the inorganic P pool in soil. Hence, the loss given by the rate of strong sorption is applied to the total inorganic P pool. The estimation of occluded P followed Wang et al. (2010) approach, based in Cross and Schlesinger (1995) the pool was assumed to be 35 % of the total soil P. $P_{leach}$ and $P_{up}$ were determined as in eq. (13), (14) based on an adaptation of Wania et al. (2012) representation of leaching and uptake of N in the new soil layer model version:

$$P_{leach} = Q_D P_{soil}, \tag{13}$$

$$P_{UP} = \sum_{PFT} \left( \frac{V_{maxp} C_{root} [P_{soil}] F_{root}}{K_{p,1/2} + [P_{soil}]} \right), \tag{14}$$

where $Q_D$ is the runoff. $V_{maxp}$ is the P maximum uptake rate, $K_{p,1/2}$ is the half saturation constant for P, $C_{root}$ is the root carbon and $F_{root}$ is the root fraction.

### 2.3.3 Organic soil phosphorus

After uptake, P is distributed in three vegetation compartments: leaf, root and wood. Leaf and root have a dynamic value that varies between a minimum and a maximum, while wood have a fix CP ratio. The vegetation P biomass dynamics is determined from the difference between the amount of uptake and the loss from litterfall as in eq. (15) and the litterfall is estimated as the CP ratio of the original model litterfall as in eq. (16):

$$\frac{dVegp}{dt} = P_{UP} - P_{LF}, \tag{15}$$

$$P_{LF} = \sum_{PFT} \frac{Lit_{leaf}}{CP_{leaf}} (1 - R_{leafp}) + \frac{Lit_{root}}{CP_{root}} + \frac{Lit_{wood}}{CP_{wood}}, \tag{16}$$

where $V_{egp}$ is the vegetation P change over time, $P_{LF}$ is the P litterfall and $Lit_{leaf}$, $Lit_{root}$, $Lit_{wood}$ are the carbon litterfall rates for vegetation carbon. The leaf CP ratio is determined as:

$$CP_{leaf} = \frac{C_{leaf}}{P_{leaf}}, \tag{17}$$

where $C_{leaf}$ is the carbon content in leafs and $P_{leaf}$ is the P content in leafs. $CP_{leaf}$ is one of the most important nutrient limitators in the model. The limiting effect of $CP_{leaf}$ is when its value is higher than the maximum $CP_{leaf}$ ratio parameter

$CP_{leafmax}$. This leads to biomass reduction. In contrast to $CN_{leaf}$, $CP_{leaf}$ does not control the maximum carboxilation rate of RuBISCO. A more detailed description of nutrient limitation can be found in section 2.4. The litter biomass is added to the P litter pool ($P_{lit}$), its dynamic is based on Wang et al. (2007) as in eq. (18):

$$\frac{dP_{Lit}}{dt} = P_{LF} - \tau_{lit} P_{lit} - P_{litmin}, \tag{18}$$

$$P_{litmin} = \frac{P_{lit}}{P_{som} + P_{lit}} P_{tase}, \tag{19}$$

$$P_{tase} = U_{tase} \frac{\lambda_{up} - \lambda_{Ptase}}{\lambda_{up} - \lambda_{Ptase} + K_{ptase}}, \tag{20}$$

where $\tau_{lit}$ is a rate constant for litter carbon decomposition (0.42 yr$^{-1}$), $P_{litmin}$ is the biochemical P litter mineralization, $P_{tase}$ is the biochemical P mineralization rate, $U_{tase}$ is the maximum rate of P biochemical mineralization, $\lambda_{up}$ is the N plant root cost to uptake P, $\lambda_{Ptase}$ is the critical value of N cost of root P uptake above which phosphate production starts and $K_{ptase}$ is the Michaelis-Menten constant for biochemical P mineralization. Here, the N cost refers to the N required for protein structures involved in the metabolization of P in plants. $P_{tase}$ is a constant value.

The soil litter decomposed is transferred to the soil organic P pool ($P_{som}$); the dynamics of $P_{som}$ are adapted from Wang et al. (2007) as in eq. (21):

$$\frac{dP_{som}}{dt} = \tau_{lit} P_{lit} \varepsilon - \tau_s P_{som} - P_{orgmin}, \tag{21}$$

$$P_{orgmin} = \frac{P_{som}}{P_{lit} + P_{som}} P_{tase}, \tag{22}$$

where the first term represents the litter P input, while the other two are the $P_{som}$ decomposition and mineralization. $\varepsilon$ is a microbial growth efficiency (0.6), $\tau_s$ is the rate constant for soil carbon decomposition and $P_{orgmin}$ is the biochemical P mineralization. The immobilization is determined from the NP ratio of the N immobilization estimated by Wania et al. (2012).

## 2.4 Nitrogen and phosphorus limitation

The N cycle limits the terrestrial vegetation productivity in two distinct ways: the first limits the photosynthesis efficiency by controlling the maximum carboxylation rate of Rubisco ($V_{cmax}$). The Rubisco enzyme plays a crucial role in the photosynthesis biochemistry by catalysing the carboxylation reactions in the Calvin cycle and has been found to be linearly related to the N leaf content (Walker et al. , 2014). The original equation for $V_{cmax}$ takes into account a fix N leaf (Cox et al. , 1999), this was

replaced by Wania et al. (2012) implemented in the first N cycle where it is replaced by the calculated inverse average canopy leaf C/N ratio ($CN_{invleaf}$), in this representation the plant productivity is reduced when $CN_{leaf}$ increases. $V_{cmax}$ is calculated as:

$$V_{cmax} = \lambda CN_{invleaf}, \tag{23}$$

where $\lambda$ is a constant of proportionality, 0.004 for C3 and 0.008 for C4 PFTs (Cox et al. , 1999). N and P both share the second form of limitation, where stoichiometrically N and P limitation reduce the vegetation biomass. If C:N ratios is too high wood and root carbon biomass is transferred to the litter pool until the normal C:N ratio is reached (See table 4).

The model assumes nutrient limitation when the estimated CN and CP leaf ratio is higher than the maximum CN ($CN_{leafmax}$) and CP ($CP_{leafmax}$) ratio in leafs. For grids with nutrient limitation the carbon in leaves is reduced to match the maximum CN or CP ratios in leafs. The carbon that is reduced is transferred to the litter pool. This reduction can happen for one or both nutrients until the ratio is met. The following equations regulate the reduction of biomass based on nutrient limitation:

$$C_{leaflimitedn} = N_{leaf} CN_{leafmax}, \tag{24}$$

$$C_{leafdiffn} = C_{leaf} - C_{leaflimitedn}, \tag{25}$$

$$C_{leaflimitedp} = N_{leaf} CP_{leafmax}, \tag{26}$$

$$C_{leafdiffp} = C_{leaf} - C_{leaflimitedp}, \tag{27}$$

where $C_{leaflimitedn}$ and $C_{leaflimitedp}$ are the carbon concentration in leafs if the system is considered to be limited. $C_{leafdiffn}$ and $C_{leafdiffp}$ are the the carbon lost due to nutrient limitation and their value are sum in the litterfall equation when the system is in nutrient limitation.

**Table 4.** Maximum leaf C:P and C:N in the CNP simulation by PFTs.

| Variables | Broadleaf trees | Needleleaf trees | C3 | C4 | Shrubs |
|---|---|---|---|---|---|
| $CP_{leafmax}$ | 225 | 250 | 500 | 500 | 450 |
| $CN_{leafmax}$ | 70 | 80 | 60 | 80 | 80 |

## 2.5 Model runs and validation

The three different terrestrial biogeochemical versions: C, CN and CNP, were run for a historical simulation from 1850 to 2020. The C version served as a baseline run representing the original version of the UVic ESCM ver. 2.10 (Mengis et al. , 2020), the CN version is the modified version of Wania et al. (2012) N model, and CNP is newest coupled model that includes P. Historical simulations are forced with fossil $CO_2$ emissions, dynamically determined land use change emissions, non-$CO_2$ GHG forcing, sulfate aerosol forcing, volcanic anomalies forcing, and solar forcing. Furthermore, 24 historical simulations were run to assess model sensitivity of 6 key parameters ($CP_{leafmax}$, $CN_{leafmax}$, $R_{leafp}$, $R_{leafn}$, $V_{maxp}$, $V_{maxn}$ ) in N and P limitation over terrestrial vegetation. The parameters were perturbed by increasing and reducing their value by 10 % and 20 % individually. $CP_{leafmax}$ and $CN_{leafmax}$ are the maximum leaf CP or CN ratios respectively. If the values of $CP_{leaf}$ and $CN_{leaf}$ are above these thresholds the model will take the system to be nutrient limited by either P or N. $R_{leafN}$ and $R_{leafP}$ are parameters that represents the resorption of N and P in leafs. This partly controls the loss of N and P from vegetation to the litter pool. $V_{maxp}$ and $V_{maxn}$ are the P and N maximum uptake rates.

It should to be noted that the porting of the N cycle from version 2.9 to 2.10 of the UVic ESCM and later model spin-up, could slightly alter the results presented in Mengis et al. (2020). Hence, our baseline model is slightly different from the standard UVIC ESCM ver. 2.10. The N cycle is compared to Zaehle et al. (2010); Li et al. (2000) and Yang et al. (2009) as well as Wania et al. (2012). The $N_2O$ flux was compared with the Emissions Database for Global Atmospheric Research (EDGAR ver. 6.0, Crippa et al. (2021)) dataset, it provides emission time series from 1970 until 2015 for non-CO2 GHGs for all countries.

For the P cycle, we used as benchmark for the carbon cycle the UVic ESCM version 2.10 model calibration values and references, which included the Le Quere et al. (2018) datasets. The total soil P was calibrated with the He et al. (2021) dataset. The labile and sorbed pools were calibrated using Yang et al. (2013) P distributions map dataset. For the use of He et al. (2021) dataset we transformed the units with Eq. 28:

$$P_{soil} = Bk_{density} * SL_D * P_{dataset}, \tag{28}$$

where $P_{soil}$ is the total P soil concentration (kg P m$^{-2}$), $Bk_{density}$ (kg m$^{-3}$) is the builk density taken from International Geosphere-Biosphere Programme Data and Information System (IGBP-DIS) (Global Soil Data Task Group , 2014), $SL_D$ (m) is the soil layer depth and $P_{dataset}$ (kg P (kg soil)-1) is He et al. (2021) dataset. The foliar stochiometry was compared to the latitudinal trend from Reich and Oleksyn (2004) N:P observations.

One of the challenges of modelling nutrients in terrestrial systems is the lack of observations and validation datasets. Furthermore, the existing range of values for N and P variables are highly uncertain. This large range in values makes it difficult to accurately tune models. Although, improvements are in sight, with new artificial intelligence derived global datasets beginning to become available (He et al. 2021). Model validation has been advancing quickly in the last decade (Spafford and MacDougall , 2021) with tools such as the International Land Model Benchmarking (Collier et al. , 2018) that significantly improves terrestrial model validation. However, there are limited variables available to compare to nutrient model develop-

ment. The increase of the addition of nutrient structures in ESMs (Arora et al. , 2020) suggest the need of terrestrial nutrient validation tools to improve model accuracy in the developmental phase. Moreover, a terrestrial nutrient model intercomparison project would unify global efforts to improve the representation of N and P in ESMs.

**Table 5.** Phoshorus cycle model pools and variables.

| Variables | Units | Descriptions |
|---|---|---|
| $P_{litmin}$ | kg P m$^{-2}$ yr$^{-1}$ | P litter mineralization |
| $P_{orgmin}$ | kg P m$^{-2}$ yr$^{-1}$ | P organic matter mineralization |
| $P_{leach}$ | kg P m$^{-2}$ yr$^{-1}$ | P leaching |
| $P_{up}$ | kg P m$^{-2}$ yr$^{-1}$ | P uptake |
| $P_{sorb}$ | kg P m$^{-2}$ yr$^{-1}$ | P sorbtion |
| $P_{imm}$ | kg P m$^{-2}$ yr$^{-1}$ | P immobilization |
| $[P_{soil}]$ | kg P m$^{-3}$ | Soil layers labile P concentration |
| $P_{soil}$ | kg P m$^{-2}$ | Labile P |
| $Lit_{leaf}$ | kg C m$^{-2}$ yr$^{-1}$ | Leaf literfall rate |
| $CP_{leaf}$ | kg C (kg P)$^{-1}$ | CP leaf ratio |
| $Lit_{root}$ | kg C m$^{-2}$ yr$^{-1}$ | Root literfall rate |
| $CP_{root}$ | kg C (kg P)$^{-1}$ | CP root ratio |
| $Lit_{wood}$ | kg C m$^{-2}$ yr$^{-1}$ | Wood literfall rate |
| $CP_{wood}$ | kg C (kg P)$^{-1}$ | CP wood ratio |
| $F_{tase}$ | kg P m$^{-2}$ yr$^{-1}$ | Rate of P biochemical mineralization |
| $P_{som}$ | kg P m$^{-2}$ | P soil organic matter pool |
| $P_{lit}$ | kg P m$^{-2}$ | P litter pool |

**Table 6.** Phosphorus cycle model parameters.

| Variables | Units | Value | Description | Source |
|---|---|---|---|---|
| $K_s$ | - | Varies with soil order | Fraction of P sorbed | Goll et al. (2017) |
| $P_{wea}$ | kg P m$^{-2}$ yr$^{1-}$ | Varies with soil order | P flux from weathering | Wang et al. (2010) |
| $\tau_{sorb}$ | yr$^{-1}$ | 0.067 | Rate of P strong soil sorption | Wang et al. (2010) |
| $K_{p,1/2}$ | kg P m$^{-3}$ | 0.002 | Half saturation constant for P uptake | Machado and Furlani (2004) |
| $V_{maxp}$ | kg P (kg root C$^{-1}$ )yr$^{-1}$ | 0.46 | Maximum uptake rate for P | Tuned |
| $R_{leaf}$ | - | 0.5 | Leaf P readsorption rate | Tuned |
| $U_{tase}$ | kg P m$^{-2}$ yr$^{-1}$ | 0.0001 | Maximum biochemical mineralization rate | Wang et al. (2007) |
| $\lambda_{up}$ | kg C (kg P)$^{-1}$ | 25 | N cost of plant root P uptake | Wang et al. (2007) |
| $\lambda_{ptase}$ | kg C (kg P)$^{-1}$ | 15 | Critical N cost of root P uptake | Wang et al. (2007) |
| $K_{ptase}$ | kg C (kg P)$^{-1}$ | 150 | Constant for biochemical P mineralization | Wang et al. (2007) |
| $\tau_{lit}$ | yr$^{-1}$ | 0.42 | Rate constant for litter C decomposition | Wang et al. (2007) |
| $\varepsilon$ | - | 0.6 | Microbial growth efficiency | Wang et al. (2007) |
| $\tau_s$ | yr$^{-1}$ | 0.02 | Constant for soil carbon decomposition | Wang et al. (2007) |
| $\lambda$ | - | Varies with PFTs | Constant of proportionality | Cox et al. (1999) |
| $CP_{leafmax}$ | kg C (kg P)$^{-1}$ | Varies with PFTs | Maximum CP ratio | Tuned |

## 3 Results and Discussions

### 3.1 Carbon cycle

#### 3.1.1 Land global primary productivity

The global gross productivity in CN and CNP resulted in a better agreement with the FLUXCOM GPP dataset (Jung et al. , 2019) as shown in Fig. 3, with both CN and CNP overestimating the terrestrial global GPP average less than the baseline simulation. Compared to the baseline simulation (143 Pg yr$^{-1}$) both nutrient limited model versions showed a reduced mean GPP from the years 2001-2015 with CN at 130 Pg yr$^{-1}$ and CNP at 127 Pg yr$^{-1}$. Furthermore, the modifications for the N cycle in regards with the mass balance changes resulted in the reduction of mean GPP from 129 Pg yr$^{-1}$ (Wania et al. , 2012) to 122 Pg yr$^{-1}$ in the 1990s. The GPP distribution from Baseline, CN and CNP reproduce FLUXCOM dataset values reasonably well (Fig. 4). The seasonal pattern of GPP is also well represented within out simulations as shown in Fig. 4. The addition of nutrients improves the representation of GPP, where CNP had the highest correlation with FLUXCOM GPP dataset. The high GPP in the baseline simulation can be explained by the overestimation of the vegetation biomass especially broadleaf trees in tropical regions stated in Mengis et al. (2020). The representation of vegetation biomass is linked to the PFTs fractions in the model. In the CN and CNP simulations the reduction of biomass is critical for the reduction of terrestrial productivity, especially in tropical regions where P availability has been shown to be a limiting factor for GPP (Du et al. , 2020). Similar to Wania et al. (2012), Bonan and Levis (2010), and Zaehle et al. (2010) the addition of nutrient limitation in ESM seems to reduce GPP. Furthermore, locally in Amazonia soils, Nakhavali et al. (2021) found that the inclusion of P reduces the model GPP and NPP outputs by 5.1 and 4.5% respectively for a site simulation. Similar to Nakhavali et al. (2021) we found an overall reduction of GPP in the Amazon region.

The nutrient limitation reduced the amount of land-atmosphere carbon flux in the simulations. The cumulative land uptake from 1850-2005 was 150 Pg C yr$^{-1}$ in CNP, lower than version 2.10 calibration in Mengis et al. (2020) (177 PgC yr$-1$). This change in response is crucial for understanding the future dynamics in the Shared Socio Economic Pathways Projections as terrestrial vegetation is expected to decrease its capacity to store carbon in the future (Goll et al. , 2012). Overall, the carbon feedback values are in concordance with the ranges of the global carbon project used in Mengis et al. (2020) (Le Quere et al. , 2018) where the cumulative carbon flux was estimated to be 141 PgC yr$^{-1}$ from 1850-2005. The atmosphere to land carbon flux follow the the GCP dataset (Le Quere et al. , 2018) magnitude closely.

Similar to Wania et al. (2012), we found higher values of NPP for CN (77.4 Pg C yr$^{-1}$) compared to the baseline simulation (74.2 Pg C yr$^{-1}$). While CNP (72 Pg C yr$^{-1}$) resulted in lower values, due to the reduction of tropical vegetation biomass. CN and CNP results are close to the upper range (21.5 to 69.3 Pg C yr$^{-1}$) of simulated NPP showed in Li et al. (2015). The reduction of tropical biomass mainly in broadleef trees carbon is reflected in the fraction of the PFT shown in the model output. Wania et al. (2012), argued that the reason behind the high NPP was the dependence of autotrophic respiration on N content in leaf, root and stem which are based on the original MOSES/TRIFFID version (Cox et al. , 1999). In CN and CNP, the reduction of wood CN ratios and higher leaf content than in CN and CNP which fluctuates from a minimum to a maximum

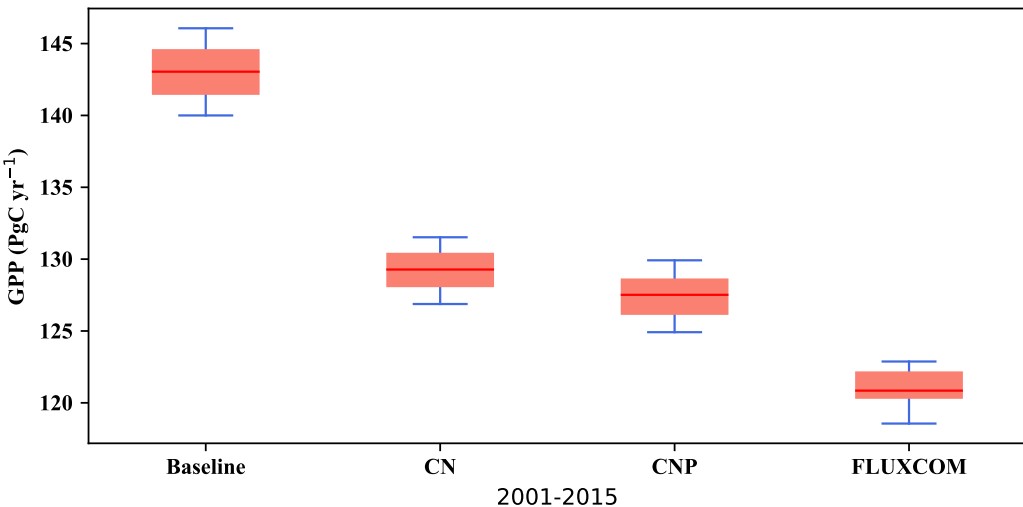

**Figure 3.** Modelled yearly Gross Primary Productivity (GPP) from 2001 to 2015 versus FLUXCOM GPP dataset (Jung et al. , 2019).

value gives place to the reduction of the maintenance respiration which reduces the autotrophic respiration and consequently
NPP. Furthermore, in the new CNP version while wood CN remains to be fixed the stochiometrical reduction of wood carbon by the lack of P availability decreases wood carbon even more especially in tropical forests and other tropical ecosystems.

### 3.1.2 Atmospheric $CO_2$ concentration

The simulated CNP atmospheric $CO_2$ concentration matches observations very closely and the addition of N and P has shown an improvement in the representation of the model accumulation of carbon in the atmosphere. The $CO_2$ concentration has
360 improved compared to the evaluated 2.10 version of the UVic ESCM where from 1960 to 2010 the simulation deviates above the observed curve ($\Delta$78 ppm in the simulation compared to $\Delta$73 ppm observations; Mengis et al. (2020)). Compared with the CN and baseline simulations (Fig. 3), CNP provides a more accurate representation of the atmospheric $CO_2$ concentration. Thus the nutrient limitation has effectively reduced the $CO_2$ fertilization effect on the terrestrial vegetation. Consequently, the CN and CNP show a larger pool of atmospheric $CO_2$.

### 365 3.1.3 Terrestrial vegetation

Given that tropical forests and savannas are commonly limited by the availability of P, the simulated vegetation biomass representation is affected by the absence of nutrient limitation in ESMs. Nakhavali et al. (2021) found that the addition of P improved the vegetation estimations and the carbon cycle response to rising $CO_2$ for the Amazonian region, basing their study in a site representative for 60% of the Amazon soils.

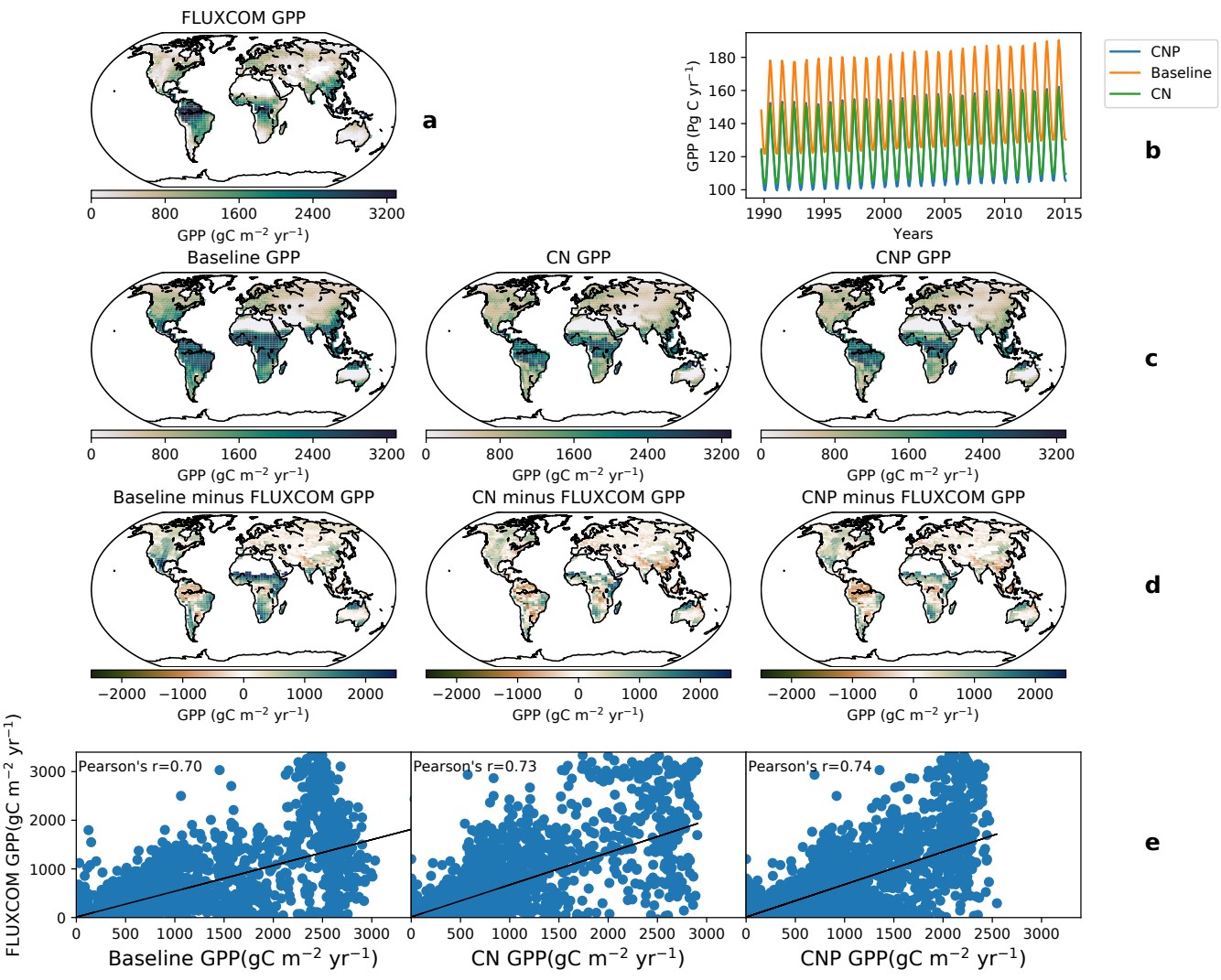

**Figure 4.** a. FLUXCOM GPP dataset from 2000-2010, b. Seasonal GPP from 1990-2015 for Baseline, CN and CNP. c. Second line shows the global GPP from 2000-2010 for Baseline, CN and CNP. d. The third line shows the difference between Baseline, CN and CNP and FLUXCOM GPP dataset. e. Shows the correlation of Baseline, CN and CNP to FLUXCOM GPP dataset.

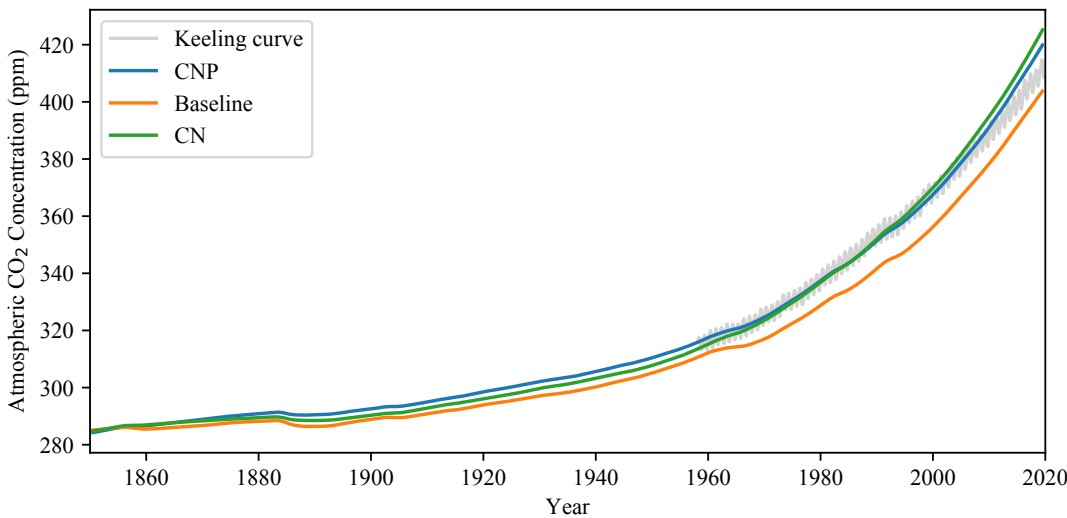

**Figure 5.** Atmospheric CO$_2$ concentration in CNP, CN and baseline simulations compared to the keeling curve from the Mauna Loa observatory (Keeling et al., 2005; grey line).

In the CNP version of the model Broadleaf trees coverage declined in tropical and subtropical latitudes (Fig. 6) with the largest changes located in South East Asia, Africa and South America. The reduction of vegetation biomass ranged from 6-20 % in South America and Africa, while a higher reduction of 20-30% was present in South Eastern Asia. The magnitude of continental difference can be attributed with the base internal vegetation biomass model version bias (Mengis et al. , 2020). Additionally, CN and CNP show a shift of coverage where broadleaf trees is taken over by C3 grass.

Needleleaf trees were reduced in North America and Europe. Both CN and CNP simulations vegetation carbon resulted in a decrease of vegetation biomass with 456 Pg C and 525 Pg C respectively compared to baseline simulation (594 Pg C), similar to Zaehle et al. (2010). Overall CNP shows a high correlation with all PFTs coverage when compared with Poulter et al. (2015) PFTs dataset. In tropical regions our model seems to represent vegetation closely to the data (Fig. 7, 8).

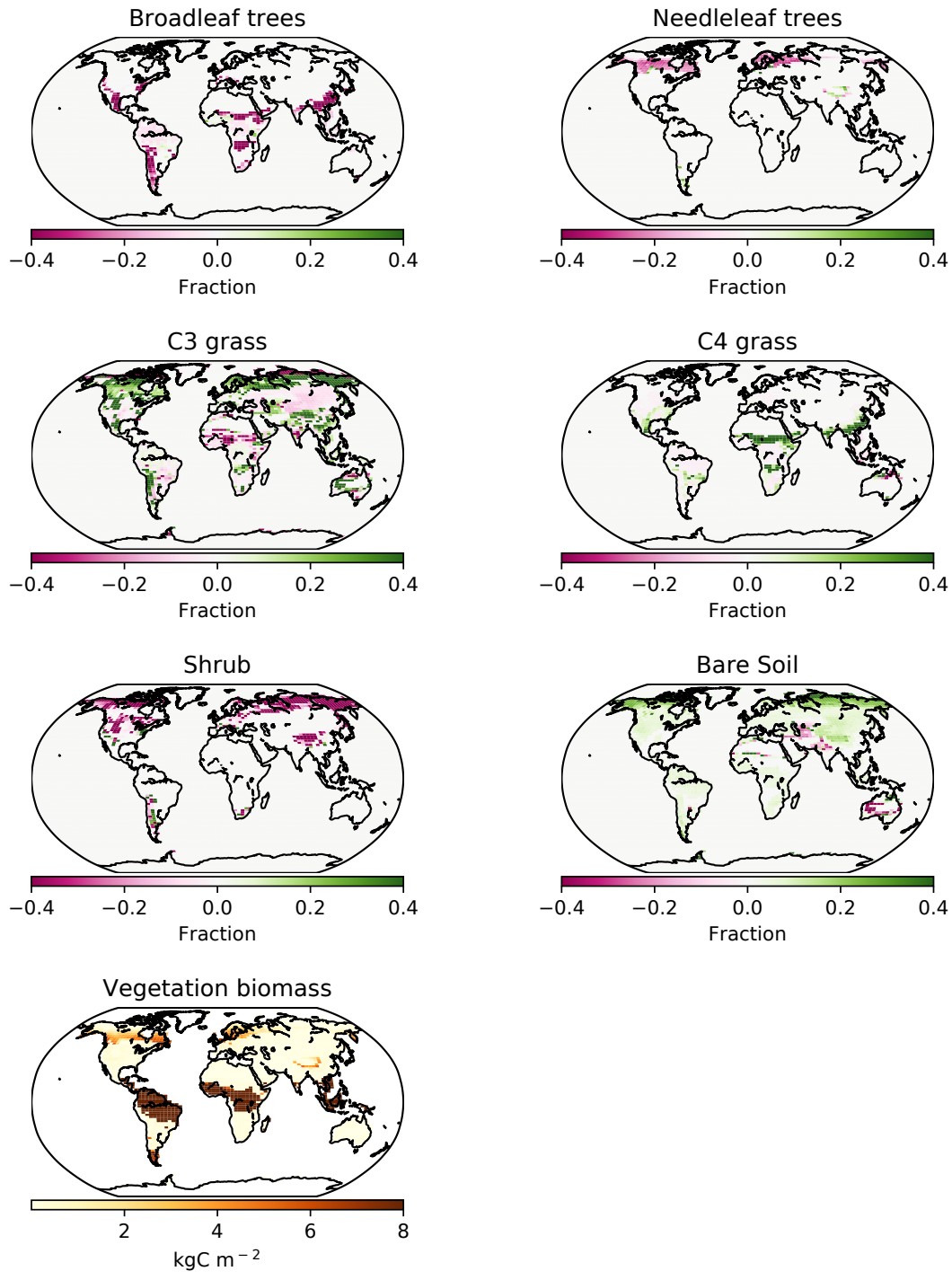

**Figure 6.** PFTs fractions in the UVic ESCM for 1980-2010, CNP minus baseline. Bottom last plot shows CNP global biomass distribution.

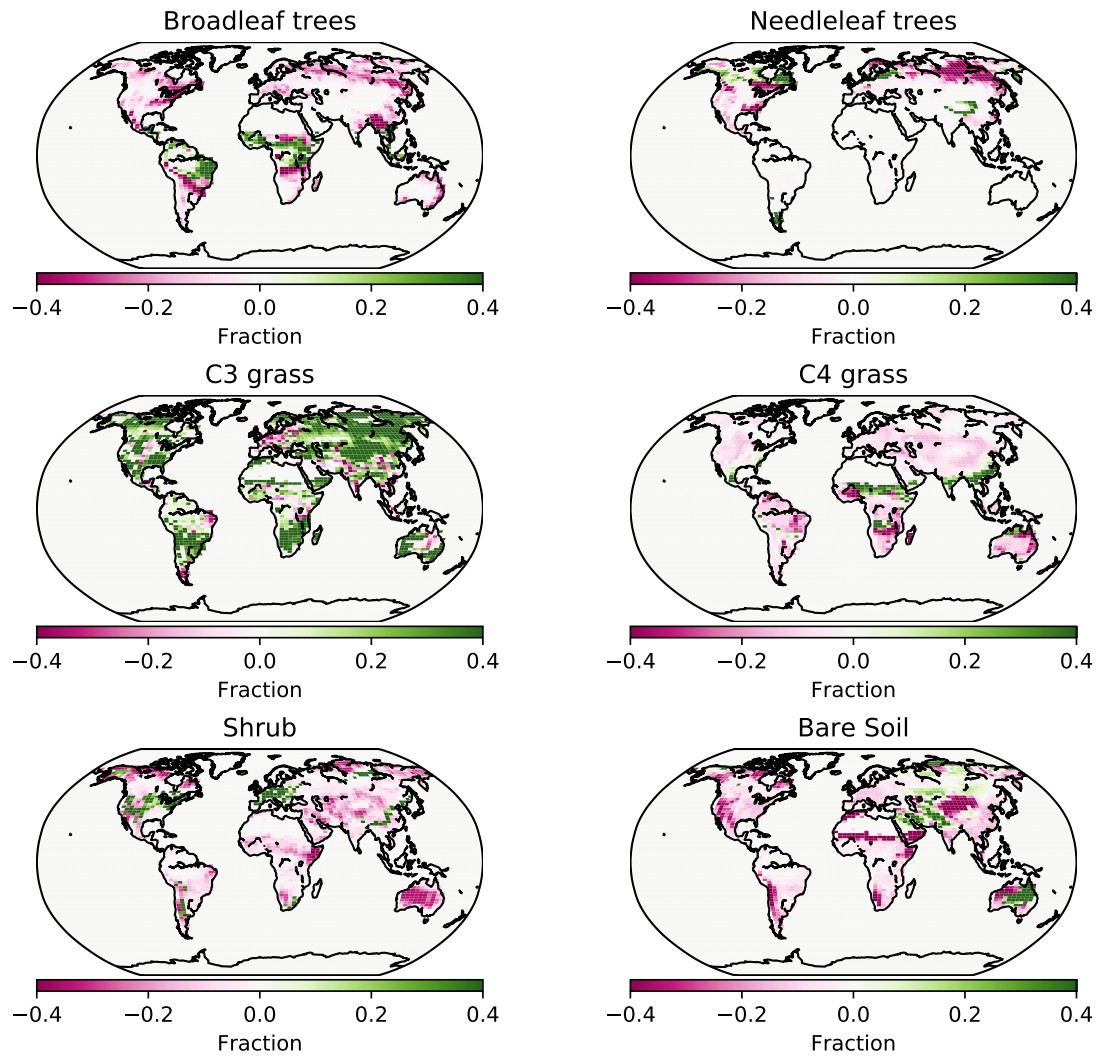

**Figure 7.** PFTs fractions in the UVic ESCM for 2008-2012, CNP minus Poulter et al. 2015 PFTs dataset.

The total vegetation carbon are similar to Wania et al. (2012), with tropical forest having a range from 8-16 kgC m$^{-2}$ and

4-12 kgC m$^{-2}$ in temperate and boreal forest with means of 10.50 and 6.7 kgC m$^{-2}$ respectively compared to 12-16 kgC m$^{-2}$

and 4-12 kgC m$^{-2}$ and means of 13.4 and 7.3 kgC m$^{-2}$. The latitudinal mean shows a decrease in the range of vegetation

carbon in tropical latitudes of 1-1.5 kgC m$^{-2}$ and 0.4-0.8 kgC m$^{-2}$ in northern template latitudes. These results indicate that

the main reduction of vegetation carbon is in the tropics, which agrees with the general N and P global pattern (Du et al. ,

2020). Consistent with Wania et al. (2012) the vegetation carbon outputs are similar to 12.1 kgC m$^{-2}$ for tropical and 5.7–6.4

for temperate and boreal forests to Malhi et al. (1999).

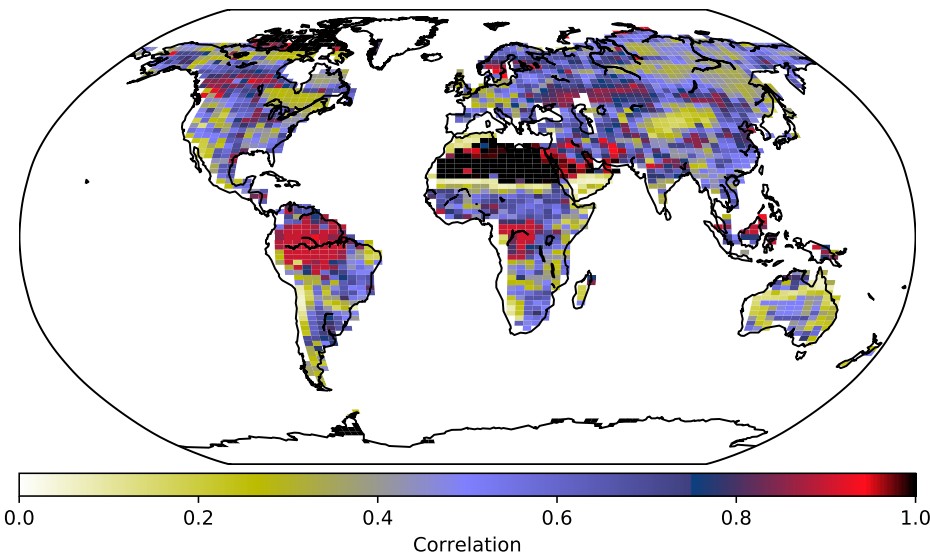

**Figure 8.** PFTs fractions across grid cells in the UVic ESCM for 2008-2012, CNP correlation to Poulter et al. (2015) PFTs dataset.

## 3.2 Nitrogen cycle

### 3.2.1 Nitrogen distribution

The soil N ranges from 0 to 1.5kgN m$^{-2}$ with lower N in tropics increasing towards the temperate regions. Globally, the CNP simulated soil N is reduced compared to the original N structure in the UVic ESCM version 2.9 presented by Wania et al.
(2012). The primary differences between Wania et al. (2012) N cycle and the current version are the soil layer structure and the stochiometry response to N limitation. In the former, N could be transfer from other pools when N was outside of the ratios threshold and thereby be considered to be limiting vegetation.

This result is also lower than the 0 to 4.8 kgN m$^{-2}$ from IGBP-DIS data base (Global Soil Data Task Group , 2000). Wania et al. (2012) stated that the N content in the model is depended to soil carbon fixed via a fixed CN ratio. Given this, lower carbon
values can lower soil N values in CN simulations. Thereby, lower carbon in soil could be a strong reason why our results have less N than IGBP-DIS data base (Global Soil Data Task Group , 2000) and Wania et al. (2012). However, our values fall within the range of uncertainty. Our model estimates a mean BNF for 2010-2020 of 119 Tg N yr$^{-1}$. This value is above 35 Tg N yr$^{-1}$ from Braghiere et al. (2022) and within the range of 52–130 Tg N yr$^{-1}$ presented by Barnard and Friedlingstein (2020)

### 3.2.2 Vegetation nitrogen

The total amount of vegetation N (2.20 Pg N) was lower than the previous N cycle (2.94 Pg N, Wania et al. (2012)). These values are similar to Zaehle et al. (2010) (3.8 Pg N) and Wang et al. (2018) (3.9 PgN) but lower than Li et al. (2000) (16Pg N) and Yang et al. (2009) (18 Pg N). Our tropical (30 to 45gN m$^{-2}$ ) and boreal forest vegetation N (20 to 35gN m-2 ) results are lower than from Wania et al. (2012) (30 to 40gN m$^{-2}$ ), and those of Xu-ri and Prentice (2008) and Yang et al. (2009) (both studies ranged between of 150 to 400 gN m$^{-2}$)

The global pattern of CN ratio is similar to Wania et al. (2012) structure with the highest located in tropical regions especially South America and South East Asia. Tropical forests show a value that ranges from 230-280 C:N (Figure 9) compared to 250-300 C:N to Wania et al. (2012). The reduction in wood carbon in tropics by P limitation in CNP lowered the C:N ratios. Our values are within the observational range of uncertainty (95-730) stated in Martius (1992).

The distribution of vegetation N resembles the results of Du et al. (2020) where N primary effect in higher latitudes. The
PFTs fraction changes show that N mainly limits North and central America (BR and NL), Chile (BR), Argentinian Patagonia (BR), North Europe (NL) and East Asia (BR)(Fig. 6). However, there seems to be N limitation in the tropical Africa and Asia in our model simulations. Even though our model does not represent co-limitation the stochiometric limitation does seems to indirectly capture this effect.

### 3.2.3 N$_2$O fluxes

The multilayer model has allowed the estimations of anoxic regions and hence, a major improvement in the model is the quantification of terrestrial N$_2$O flux. Figure 10 shows CN and CNP N$_2$O fluxes from 1990 to 2018. Compared to EDGAR version 6 dataset (Crippa et al. , 2021) our model simulates N$_2$O fluxes relatively well, agreeing mostly in last 10 years of the values. However, we observed an overestimation from 1990 to 2010. The CN version of the model fit within the lower natural (Natural soil, Atmospheric N deposition on land) + anthropogenic (Agriculture, Fossil fuel and industry) emission
range (8.9 -14.3 Tg N yr$^{-1}$) given by the global carbon project (Tian , 2020) while CNP fall just below the lower range value. The reduction of N in the model system by P effect is shown by this results, the reduction of vegetation biomass and then litterfall reduces the amount of N transfer to the N soil pool limiting the natural denitrification. The lack of oceanic production of N$_2$O in the model makes the comparison with the global total N$_2$O flux impossible at the moment. The total estimates for N$_2$O emissions being 4.2 to 11.4 Tg N yr$^{-1}$ anthropogenic and 8.0 to 12.0 Tg N yr$^{-1}$ natural given by global carbon project
(Tian , 2020). Assuming an ocean output of a mid-range emission (3.4 Tg N yr$^{-1}$) the model simulations are close to the lower range of the emission reported with CN (13.3 Tg N yr$^{-1}$) and CNP (12.1 Tg N yr$^{-1}$). Lighting and atmospheric production, biomass burning (addition of N$_2$O to atmospheric pool) and post deforestation pulse effect are not taken into account in the model structure and that could improve the fit of the simulation to a mid-range level value.

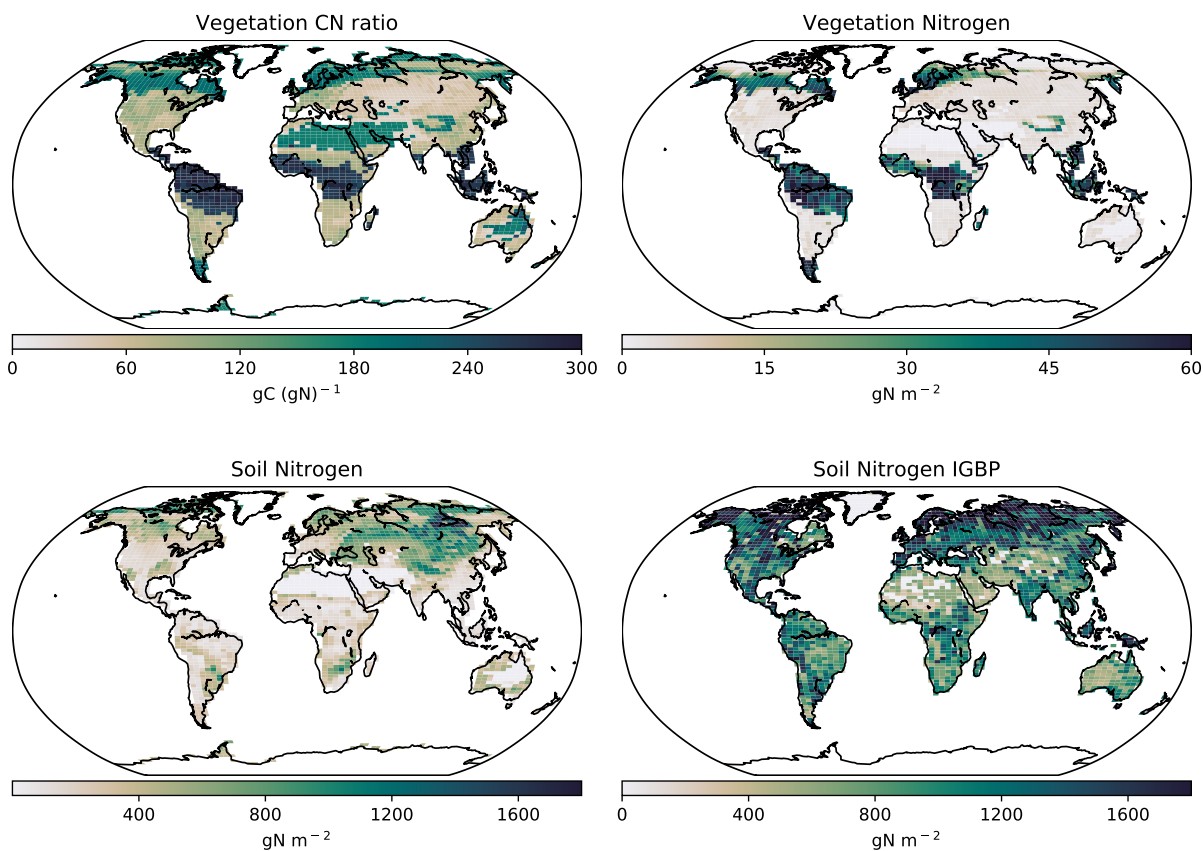

**Figure 9.** Modelled global soil and vegetation N in the CNP version of the UVic ESCM from 1980-1999. Lower right map corresponds to the soil N from the IGBP-DIS dataset (Global Soil Data Task Group , 2000)

## 3.3 Phosphorus cycle

### 3.3.1 Inputs and losses

The P global weathering rate estimated is 3 Tg P yr$^{-1}$ similar to 2 Tg P yr$^{-1}$ in Wang et al. (2010). Fertilization inputs of 1 Tg P yr$^{-1}$ (Filipelli , 2002) where added as an option to the model but were not used for the current simulations and dust deposition is not accounted for. Hence, the only P input into the system in this experimental set-up comes from rock weathering. Regarding the P weathering representation Hartmann et al. (2014) approach was tested at first, but Wang et al. (2010) weathering scheme resulted in a better, simplified and controllable input. Although, Hartmann et al. (2014) was found

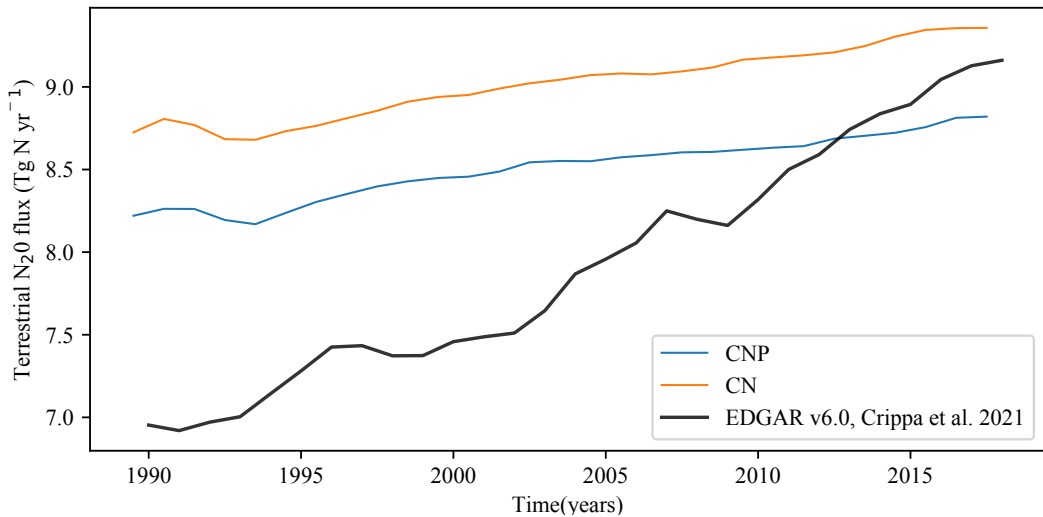

**Figure 10.** CNP and CN global soil N$_2$O emissions vs EDGAR version 6.0 N$_2$O dataset (Crippa et al. , 2021).

to be superior since P input is dynamic, incorporating model runoff and lithological map distribution. A dynamic P input will also require a better representation of P losses in order to maintain a steady state.

The P weathering was set so the loss by leaching (3 Tg P yr$^{-1}$) were in accordance with the rates of was comparable with to the riverine input stated in Filipelli  (2002) of 4-6 Tg P yr$^{-1}$. The gap corresponds to anthropogenic inputs not included here, the pre-industrial P input to the ocean from riverine input is 2-3 Tg P yr$^{-1}$ and the human activities especially agriculture (fertilizers) and water wastes roughly correspond to a doubling of the P input.

### 3.3.2   Land P pools and storages

The total inorganic and organic P values are similar as those shown in the results of Smil  (2000), Mackenzie  (2002) and Wang et al.  (2010) (Table 7), although organic P is slightly underestimated in the model (3.5 Pg P). This underestimation is likely the result of the lack of P fertilization on land. The labile, sorbed, strongly sorbed P and occluded pools are comparable values to Wang et al.  (2010).

Globally the total soil P distribution (Fig. 11) is comparable to the He et al.  (2021) dataset, which is one of the few terrestrial P concentrations maps available. Overall, the model simulates less global P especially in northern latitudes most likely due to the oversimplified weathering scheme that underestimated the inputs in higher latitudes.

Latitudinally, the tropical soils showed the lowest P with exception of highlands and mountains while increasing sequentially to the northern latitudes as showed in He et al.  (2021). The labile P shows a similar distribution to Yang et al.  (2013) with tropical regions being relatively depleted compared to other regions due to the high adsorption and occlusion by the soils.

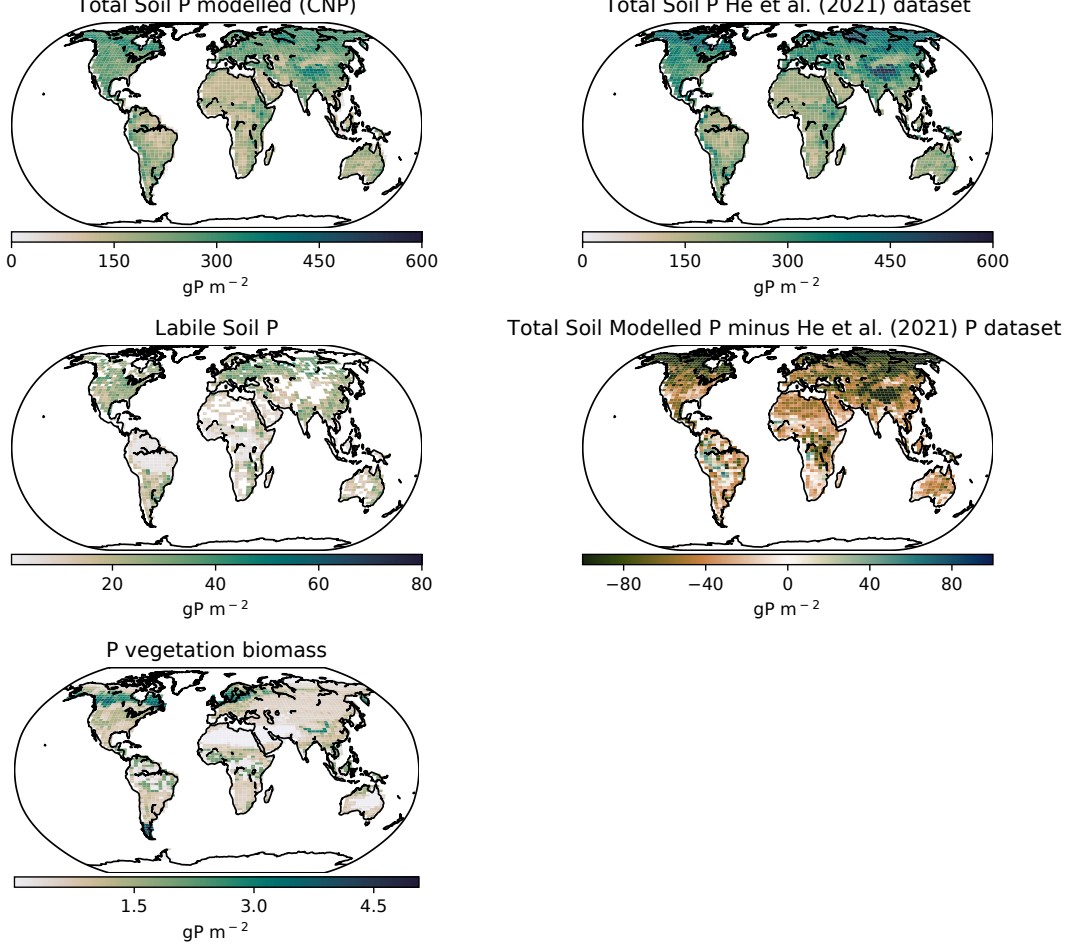

**Figure 11.** Soil and vegetation P global distribution. Modelled total P in soil, total P in soil as in He et al. 2021, soil P, labile P, vegetation biomass and the difference between modelled and observational P from He et al. (2021).

In contrast with N, P inputs is limited by the mineral (apatite) concentration and weathering rate rather than biologically fixed. Most of the P is retain by soils leaving a small labile fraction for biological uptake. Because P mineral weathering and chemical recycling in the soils are so constraining, our linear model approach for adsorption based on Goll et al. (2017) might overestimate the impact of adsorption and occlusion in tropical soils. It is also worth noting that the biological impact on the adsorption-desorption dynamics is missing in most P modules in ESMs. The release of P from mineral grains can be enhanced by either the reduction of pH due to respiration, the direct addition of organic acids by plants roots Schlesinger (1997).

### 3.3.3 Phosphorus in vegetation

The terrestrial vegetation shows a slight underestimation in comparison with other models. The new stochiometry limitation scheme of the model plays an important role in the vegetation biomass and could be the reason for the underestimated values specially for tropical regions. However, the range of P in terrestrial vegetation is still uncertain with several studies showing a range from 1.8-3.0 Pg P (Smil , 2000). Although Wang et al. (2010) have dismissed those values as overestimations given an overall N:P ratio of 10-20 gN gP $^{-1}$, 3 Pg P is simply too high to be met.

**Table 7.** Phosphorus cycle model pools and values for literature.

| Variables | Value (Pg P ) | References (Pg P ) |
|---|---|---|
| Total inorganic P | 20.8 | 35-40 Smil (2000) |
| Total organic P | 3.5 | 36 Mackenzie (2002) |
| | | 26.5 Wang et al. (2010) |
| | | 13.7 Wang et al. (2018) |
| | | 5-10 Smil (2000) |
| Labile P | 1.4 | 5 Mackenzie (2002) |
| | | 5.7 Wang et al. (2010) |
| | | 8.6 Yang et al. (2013) |
| | | 1.5 Wang et al. (2010) |
| Sorbed P | 1.1 | 3.6 Yang et al. (2013) |
| | | 1.7 Wang et al. (2010) |
| Strongly sorbed P | 12 | 7.6 Wang et al. (2010) |
| Occluded | 6.3 | 9.0 Wang et al. (2010) |
| Vegetation P | 0.2 | 0.4 Wang et al. (2010) |
| | | 0.5 Smil (2000) |
| | | 0.5 Wang et al. (2018) |
| | | 0.2 Wang et al. (2018) |
| P Litter | 0.01 | 0.5 Mackenzie (2002) |
| | | 0.04 Wang et al. (2010) |
| | | 0.03 Wang et al. (2018) |

The foliar stochiometry seems to approximately follow the N:P ratio field measurements of Reich and Oleksyn (2004) (Fig. 12). The tropical regions show some underestimated values in our model, the low amount of labile P and the latter decrease in broadleafs trees biomass could be responsible for the low numbers. Similarly, Nakhavali et al. (2021) show model values of 4-15 gP m$^{-2}$ for an Amazonian site which surpasses our results.

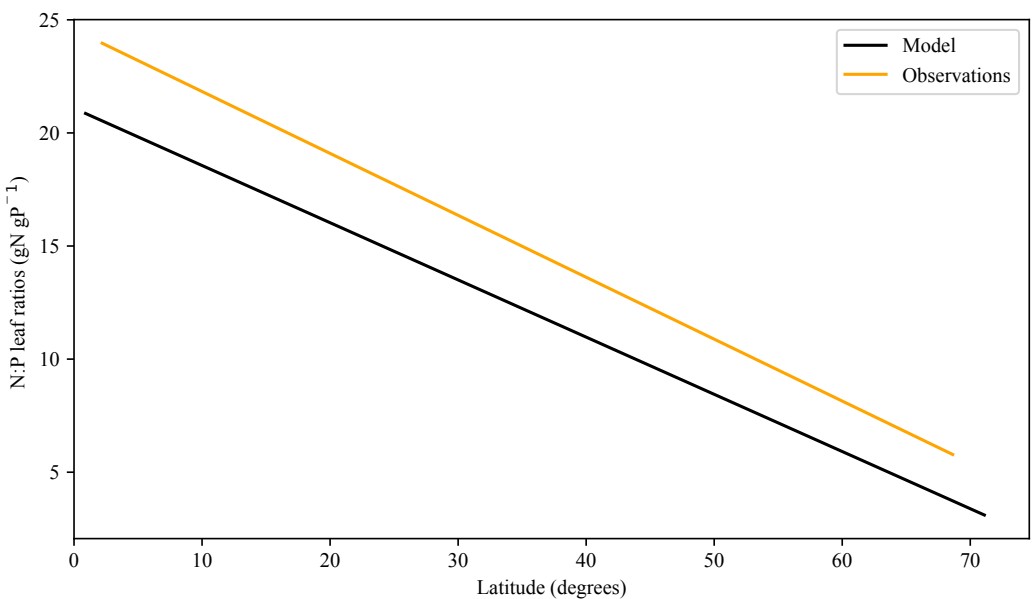

**Figure 12.** Modelled N:P leaf ratios trend vs an empirical relationship derived from Reich and Oleksyn (2004).

A more complex adsorption – desorption scheme might be beneficial to solve the underestimation for tropical latitudes as
those regions are heavily sorbed and loose most of the input P, even though, the need of a proper global P vegetation dataset
is imperative to have proper ranges in global distributions. The mechanical reduction of vegetation stochiometrically by the
model structure might also be too simplistic to represent P limitation in tropics.

### 3.4 Parameter sensitivity

We perturbed 6 parameters ($CP_{leafmax}$, $CN_{leafmax}$, $R_{leafp}$, $R_{leafn}$, $V_{maxp}$, $V_{maxn}$ ) over historical simulations to assess the
475 model sensitivity in terms of limitation of N and P. All of the above parameters play an important role in the nutrient limitation
structure of the model. $P_{leafmax}$, $N_{leafmax}$ control when the stochiometrical limitation is set to be enforced on terrestrial
vegetation and $R_{leafp}$, $R_{leafp}$, $V_{maxp}$ and $V_{maxn}$ control the uptake, litterfall and allocation of nutrients in leafs. In each
case, default values were increased and decreased by 10% and 20% while holding other parameters constant. The results were
compared to model simulations with all parameters held constant and set to default values. The cumulative atmosphere-land
carbon flux was used to measure the effect of the perturbation, since the limitation directly affects this flux.

The results of the sensitivity study show that model sensitivity varies with different parameters (Table 8). The UVic ESCM is
most sensitive to perturbations of $CP_{leafmax}$ and $CN_{leafmax}$ because both determine directly the threshold by which vegetation
carbon is reduced and nutrient limitation is defined. The model seems to be most sensitive to changes in $CP_{leafmax}$. The

regulation of this parameter is very useful to calibrate woody vegetation in tropical regions to improve the cover representation.
The other parameters have a lower impact on the atmosphere-land carbon flux ranging from -3.23% to +1.60%.

**Table 8.** Cumulative atmosphere-land carbon flux anomaly from baseline (%). The parameters were perturbed by increasing and reducing 10 and 20 % of their value.

| Parameters | -20% | -10% | +10% | +20% |
|---|---|---|---|---|
| $CP_{leafmax}$ | -16.04% | -3.03% | +0.25% | +0.26% |
| $CN_{leafmax}$ | -6.46 % | -2.10 % | +8.63% | +12.58% |
| $R_{leafp}$ | -0.23% | -0.12% | +0.22% | +0.26% |
| $R_{leafn}$ | -0.98% | -0.76% | +1.20% | +1.60% |
| $V_{maxp}$ | -3.23% | -0.94% | +0.18% | +0.22% |
| $V_{maxn}$ | -1.30% | -1.10% | +0.95% | +1.45% |

## 4 Limitations and applications of the terrestrial nutrient modules

The UVic ESCM has been a critical tool in developing the cumulative emissions framework to climate mitigation (Zickfeld et al. , 2009; Matthews et al. , 2009; Matthews and Weaver , 2009; MacDougall and Knutti , 2016; Mengis et al. , 2018; Tokarska et al. , 2019) due to its low computations cost and strict enforcement of matter and energy conservation the model is capable
of conducting a host of simulation beyond the limits of most other models, but with higher resolution than other EMICs (e.g. Montenegro et al. (2007); Matthews and Caldeira (2008); Keller et al. (2014); MacDougall and Knutti (2016); MacDougall (2017); Pahlow et al. (2020); Kvale et al. (2021)) . As terrestrial nutrient limitation constrains the carbon cycle in nature, the new N and P modules allows addressing research questions relating carbon budgets, carbon cycle and $CH_4$ feedbacks, carbon dioxide removal and permafrost carbon cycle, among other questions. Furthermore, the N and P cycles can represent
environmental and climate critical processes such as the release of $N_2O$, agricultural impacts on terrestrial soils and coastal lines, eutrophication, anoxic events and nutrient fluxes from land to ocean.

A number of limitations have been identified with the developed N and P modules that relate to the degree of complexity or the lack of large-scale datasets available. Due to the lack of global estimates of nutrient pools and fluxes based on field measurements, many of the parameters or parameterizations in this model are poorly constrained. In general, these are the
500 following model limitation that are plan to be improved in future model development projects:

1. The model does not include a dynamic nutrient leaf resorption rate. Under nutrient limitations, this rate can increase as a strategy to conserve nutrients (Reed et al. , 2012). Thus, the effect of limitation in our model might be overestimated.

2. There is a static input of P from weathering. To control the P input we chose to estimate weathering flux by adding a fix amount. This oversimplification could add more uncertainty to the P pools and can be overcome using a runoff based
weathering scheme. Moreover, we do not account for P atmospheric dust deposition.

3. The sorption-desorption dynamics of P in soil are oversimplified. We chose Goll et al. (2017) approach because it was a simpler way to represent this process. However, a more complex solution might improve the distribution P globally.

4. The absence of an ocean $N_2O$ output. Consequently, we are unable to estimate the total amount of a dynamically evolving $N_2O$ concentration at this time. As $N_2O$ is the 3rd most important greenhouse gas (IPCC , 2022), its incorporation into the model is a priority.

5. The model does not account for root uptake constrains of N and P on terrestrial vegetation. This includes spatial representations of mycorrhizal associations and the carbon cost of nitrogen and phosphorus uptake from soil (Shi et al. , 2016; Braghiere et al. , 2021, 2022).

The CNP model is primarily designed to improve carbon cycle feedbacks under current and future climate conditions. The use of nutrient limitation improves the land-atmosphere dynamics. In simulations, this improvement has a significant impact on atmospheric $CO_2$ concentrations. In future studies, we intend to assess the impact of nutrient limitation on different SSP scenarios and key carbon cycle benchmark metrics. Furthermore, the model can be used to improve the vegetation representation in ESMs. Finally, the CNP model may be used to generate coastal nutrient input and to integrate terrestrial nutrient biogeochemical processes with oceanic processes.

## 5 Conclusion

The N and P cycles simulated here fit the range of uncertainty shown in data-sets and other modelling efforts. Generally, our values fall into the lower range of the spectrum. N limits mainly high latitudes especially in northern regions, but do shows some limitation in tropical Africa and Asia. P limitations is greater in tropical regions and reduced the vegetation biomass compared to the carbon only version of the model bringing the model closer in line with observation (Mengis et al. (2020)).

The two nutrient limitation have improved the representation of the atmospheric carbon concentration in simulations forced with $CO_2$ emissions, using the Keeling curve as benchmark data. The land-atmospheric flux fits other simulations datasets and have been reduced from Mengis et al. (2020) values. Overall N and P addition have improved the carbon cycle feedbacks simulated in historical simulations. The GPP is lowered especially in tropics mainly due to the reduction of woody vegetation biomass.

Many improvements remain to be made in our model structure. In regards with N cycle denitrification processes need to be improved, $N_2O$ fluxes while in the same magnitude as observations lack the trend showed in other benchmark datasets. The complexity of the P cycle could be improved especially the input and sorption processes. Finally both N and P cycles could gain accuracy from adding dynamics leaf re-absorption rates that has been shown to change when nutrient limitation is present in the ecosystem and that can be used as in Du et al. (2020) to clearly map the limitation pattern. Despite these limitations the improved model has shown higher fidelity to observations and is expected to improve projections of the future of key carbon cycle feedbacks.

*Code and data availability.*

The model code and data used for this paper are available at:
https://borealisdata.ca/dataset.xhtml?persistentId=doi:10.5683/SP3/GXYZKU.

*Author contributions.*

AHMD initiated the redevelopment of the N module. MD ported the N module to the UVic ESCM version 2.10 and further improved it. MD developed the P cycle. MD wrote the paper and AHMD provided supervisory support. NM contributed with the interpretation and validation of the model results. SA contributed with the visual representation of the P module structure and model results.

*Competing interests.*

The authors declare that they have no conflict of interest.

*Acknowledgements.* AHMD and MD are grateful for support from the Natural Science and Engineering Research Council of Canada Discovery Grant program and support from Compute Canada (now the Digital Research Alliance of Canada). We are indebted to M. Eby for early advise on implementing the Nitrogen version of the model and for providing the model code for the original N cycle verion of UVic
ESCM.

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
