# Peer review of "Modelling the terrestrial nitrogen and phosphorus cycle in the UVic ESCM"

_Geoscientific Model Development, 2022_

## Author Comment (AC1)

**Response to reviewers**

We appreciate the time and effort that the editors have dedicated to providing your valuable feedback on our manuscript. The reviews are copied verbatim and are italicized. Author responses are in regular font. Changes made to the manuscript are blue.

**Comments from reviewer 1**

**General Comment**

*This manuscript describes new nitrogen (N) and phosphorus (P) cycling processes that have been added to a new version of the UVic ESCM intermediate complexity Earth system model. The new processes are well justified, clearly described and well supported by citations to relevant literature, data syntheses, and theory. The new model functions are evaluated using global datasets of carbon, nitrogen, and phosphorus pools and/or fluxes and the level of agreement and areas for improvement in the model are described clearly and fairly. Overall, this is a well-written model description paper.*

**Response**

Thank you for the positive feedback.

**General Comment**

*There are some areas where the clarity of the manuscript could be improved, primarily related to the equations and figures. While the Methods section does include several relevant equations for N and P cycling, it omits some important processes and overall does not provide a complete picture of N and P cycling in the model. Importantly, equations and explanations are not provided for the variable tissue C:N and C:P ratios that are an important part of the stoichiometric limitation component of the model. I would advise including those equations in the main text since they are a key part of the model functionality and*

*results. I would also suggest adding an appendix with the complete set of equations related to N and P*

*cycling so readers do not need to search through other previous papers to gain a complete picture of how*

25 *nutrient cycling in the model works.*

**Response**

Thank for your comment. The equations governing C:N and C:P ratios were added in the manuscript in section 2.2, as well as a more in depth review on the limitation processes in section 2.4.

Section 2.2:

$$CN_{leaf} = \frac{C_{leaf}}{N_{leaf}}, \tag{1}$$

where $C_{leaf}$ is the carbon content in leafs and $N_{leaf}$ is the nitrogen content in leafs. $CN_{leaf}$ is one of the most important nutrient limitators in the model. It controls the maximum carboxilation rate of RuBISCO. Furthermore, it control vegetation biomass. If nitrogen concentration in leafs is higher than $CN_{leafmax}$ (the maximum CN ratio parameter) terrestrial vegetation biomass is reduced.

35 Section 2.4:

The model assumes nutrient limitation when the concentration of nitrogen and phosphorus is higher than the maximum CN ($CN_{leafmax}$) and CP ($CP_{leafmax}$) in leafs. For grids with nutrient limitation the carbon in leaves is reduced to match the maximum CN or CP ratios in leafs. The carbon that is reduced is transferred to the litter pool. This reduction can happen for one or both nutrients until the ratio is met.

40 The following equations regulate the reduction of biomass based on nutrient limitation:

$$C_{leaflimitedn} = N_{leaf}CN_{leafmax}, \tag{2}$$

$$C_{leafdiffn} = C_{leaf} - C_{leaflimitedn}, \tag{3}$$

45 $$C_{leaflimitedp} = N_{leaf}CP_{leafmax}, \tag{4}$$

$$C_{leafdiffp} = C_{leaf} - C_{leaflimitedp}, \qquad (5)$$

where $C_{leaflimitedn}$ and $C_{leaflimitedp}$ are the carbon concentration in leafs if the system is considered to be limited. $C_{leafdiffn}$ and $C_{leafdiffp}$ are the the carbon lost due to nutrient limitation and their value are sum in the litterfall equation when the system is in nutrient limitation.

**General Comment**

*The figures are generally informative, but there were some parts of the text that described model-data comparisons and other patterns that were not directly shown in the figures. In addition, I think it would improve the readability of the paper if the figures showing N results matched the figures showing P results in their content. Currently, the N figures and P figures show different comparisons in some cases which makes it less straightforward for readers to evaluate those parts of the model. For example, Figure 9 shows maps of modeled total soil P compared with measured total soil P and the difference between them. In contrast, Figure 7 shows maps of modeled N but does not show any direct comparison with measured patterns of N, even though model-data comparisons of global N patterns are discussed in the text.*

**Response**

Thank you for your comment. The figures show different comparisons in order to be compatible for comparison with other studies. It is correct that Figure 7. does not show any direct comparison. We have now added a nitrogen soil distribution map from The Global Soil Data Task.

[Figure]

**Figure 1.** Modelled global soil and vegetation nitrogen in the CNP version of the UVic ESCM from 1980-1999. Lower right map corresponds to the soil nitrogen from the IGBP-DIS dataset (Global Soil Data Task Group , 2000)

**Specific comments**

**Comment 1**

*Line 14: It would be helpful to include the GPP for carbon-only simulations in this comparison as well.*

**Response 1**

The text has been changed to:

For the years 2010-2020 the nutrient limitation resulted in a reduction of GPP from the Carbon-Nitrogen model version of 133 Pg yr$^{-1}$ and the Carbon-Nitrogen-Phosphorus model version of 129 Pg yr$^{-1}$ simulations compared to Carbon only value of 143 Pg yr$^{-1}$. This implies that the model efficiently represents a nutrient limitation over the $CO_2$ fertilization effect.

**Comment 2**

*Line 59-61: This list was hard to follow and could use some editing.*

**Response 2**

Thank you for your comment. The sentence was re-written to:

by the geochemical interactions in terrestrial soils, Vitousek et al. (2010) defined six mechanisms by which P is driven to limitation: loss by leaching, soil barriers that physically prevents access to roots, slow release of mineral P forms, P parent material, sequestration of P in soils and pools in the ecosystem and finally, anthropogenic input of nutrients.

**Comment 3**

*Line 69-75: The introduction discusses Earth system models in general, and then the history of UVic ESCM in particular). I think it would be helpful to include a few sentences about intermediate complexity*

*ESMs as a class to provide some more context about the goals of the type of model that UVic ESCM represents and how it compares to other similar models.*

**Response 3**

Thank you for the comment. We have now included a few sentences about intermediate complexity ESMs.

90

Intermediate complexity Earth system models, have a lower spatial representation, and model structures that have been intentionally simplified in one or more ways. This simplification allows for long-term simulations that are typically not feasible in higher complexity models. This class of model is not suitable for studying processes at small spatial scales. Hence, they are used in research questions that require

95  large spatial and temporal scales (Weber , 2010). The current generation Earth system models are, or have already, developed nutrient limitation for their model structure. While CN models are more common CNP models remain rarer. However, P cycles have been suggested to be included into Earth system model for its importance in tropical regions (Wang et al. , 2010; Goll et al. , 2012). The first attempt to include nutrient limitation in the University of Victoria Earth system and climate model (UVic ESCM) was done

100  by Wania et al. (2012) but was not included in the current publically available version of the model due to the need of further improvement. Hence, here we intent to improve the current state of the previous N cycle, develop a new P cycle and couple CNP in the UVic ESCM, in order to improve the carbon feedbacks projections.

**Comment 4**

105  *Line 105: Variable C:N ratios for leaf and root pools are mentioned but the details (and equations) of what determines the actual C:N ratio are not provided. It would also help to explain here how the variable C:N ratios affect other parts of the model (e.g., photosynthesis or root function dependence on tissue N). It would help to provide some summary of how the relevant processes from the Gerber et al citation are calculated, ideally with equations provided in an appendix.*

 **Response 4**

Thank you for you comment. The following equation and text were added to section 2.2.

$$CN_{leaf} = \frac{C_{leaf}}{N_{leaf}},$$ (6)

where $C_{leaf}$ is the carbon content in leafs and $N_{leaf}$ is the nitrogen content in leafs. $CN_{leaf}$ is one of the most important nutrient limitators in the model, as it controls the maximum carboxilation rate of RuBISCO. Furthermore, it control vegetation biomass. If nitrogen concentration in leafs is higher than $CN_{leafmax}$ (the maximum CN ratio parameter) terrestrial vegetation biomass is reduced. A more detailed description of nutrient limitation can be found in section 2.4.

**Comment 5**

*Equations 1-2: An explanation should be provided here for what "av" means in the mineral nitrogen pools. Later in the paper I found that this means "available" but that should be explained here, along with an explanation of how the available fraction is calculated. Is there an unavailable fraction?*

**Response 5**

Thank you for your comment. An explanation has been added to the manuscript:

where $NH_4^{UP}$ and $NO_3^{UP}$ represent the nitrogen uptake, the left term is the active uptake while the right term is the passive uptake (see table 1), the latter is the transport of N via the transpiration water stream. $Vmaxn$ is the maximum uptake rate for nitrogen, $Croot$ is the root carbon biomass, $[NH_4(av)]$, $[NO_3(av)]$ and $[Nmin(av)]$ are the $NH_4$, $NO_3$ and mineral nitrogen concentrations, $K_{n,1/2}$ is the half saturation constant for nitrogen and $Qt$ is the transpiration rate. $av$ represents the available portion of $NH_4$ and $NO_3$ in soil. This fraction is calculated as the total concentration of $NH_4$ and $NO_3$ divided by sorption factors (10 and 1 respectively) following Wania et al. (2012).

**Comment 6**

*Line 122-123: I found "depth of soil layer" and "root depth" confusing. Is the soil layer referring to each individual model layer, or to the depth of the entire soil? Is the root depth a rooting depth parameter for each PFT, or the depth at which the root fraction is being calculated?*

**Response 6**

Thank you for your comment. Depth of soil layer is the depth of each of the eight active soil layers. The UVic ESCM version 2.10 contains 14 subsurface layers with thickness exponentially increasing with depth with a surface layer of 0.1 m, a bottom layer of 104.4m and a total layer of 250 m. Only eight are active in the water and biogeochemcial cycles, the remaining layers are a granite-like heat sink.

For this equation, depth of soil layer represents the depth of each specific soil layer. Root depth is a PFT based parameter that represents the depth of the roots. In the UVic ESCM, this was first developed in a doctoral thesis written by Avis (2012).

where $Z_{top}$ and $Z_{bot}$ represents the top layer and bottom layer depth respectively, while $D$ and $dr$ are the depth of the soil layer and the root depth. The depth of soil layer represents the depth of each specific soil layer. Root depth is a PFT based parameter that represents the depth of the roots. Given the multiple soil layer set up, the root fraction modifies the value of root carbon, creating a more realistic representation of the uptake root depth reach for each PFT given the multiple soil layer set up.

**Comment 7**

*Line 129: Provide an explanation or reference for the statement that "It takes 1 mol of NO3 to mineralize 1 mol of organic C."*

**Response 7**

Thank you for your comment. We have realised that the statement only applies for very specific respiration pathways. Hence, we have deleted it from the text.

**Comment 8**

*Line 132-133: The temperature and moisture functions are not provided or explained. Is a moisture function necessary when the anaerobic respiration is calculated only for the saturated fraction of the layer?*

**Response 8**

You are correct the temperature and moisture function are not defined here. They were originally designed in TRIFFID by Cox (2001). The following equations and explanation has been added:

The temperature and moisture soil functions are taken directly from Cox (2001), and are represented by the following equations:

$$f_t = q_{10}^{0.1(t_s - 25)},  \tag{7}$$

$$f_m = \begin{cases} 1 - 0.8(S - S_0) & \text{for } S > S_0, \\ 0.2 + 0.5(\frac{S - S_W}{S_0 - S_W}) & \text{for } S_W < S \leq S_0, \\ 0.2 & \text{for } S \leq S_W, \end{cases}  \tag{8}$$

where in $f_t$, $q_{10} = 2$ and $t_s$ is the soil temperature in °C. In $f_m$, S is the soil moisture, $S_W$ is the wilting point of soil moisture, $S_0$ is the optimum soil moisture.

**Comment 9**

*Table 1: Descriptions should include "pool" or "rate" or similar for each line since the table contains a mix of different types. Also, temperature and moisture functions are functions rather than numbers or outputs and feel out of place in this table.*

**Response 9**

175 The table has been modified to:

**Table 1.** Updated nitrogen cycle module pools, rates and variables.

| Variables | Units | Type | Descriptions |
|---|---|---|---|
| $NH_4^{UP}$ | kg N m$^{-2}$ yr$^{-1}$ | Rate | $NH_4$ vegetation uptake |
| $NO_3^{UP}$ | kg N m$^{-2}$ yr$^{-1}$ | Rate | $NO_3$ vegetation uptake |
| Croot | Kg C m$^{-2}$ | Pool | Root carbon |
| $[NH_4(av)]$ | kg N m$^{-3}$ | Pool | Available $NH_4$ concentration |
| $[NO_3(av)]$ | kg N m$^{-3}$ | Pool | Available $NO_3$ concentration |
| Froot | - | Variable | Root fraction |
| $[Nmin(av)]$ | kg N m$^{-3}$ | Pool | Available mineral N concentration |
| $R_{an}$ | kg C m$^{-3}$ s$^{-1}$ | Rate | Anaerobic respiration rate |
| $C_s$ | kg C m$^{-3}$ | Pool | Density of soil carbon in each layer |
| $A_f$ | - | Variable | Anaerobic saturation fraction |
| $N_2O$ | kg N m$^{-2}$ yr$^{-1}$ | Rate | Nitrous oxide flux |
| NO | kg N m$^{-2}$yr$^{-1}$ | Rate | Nitric oxide flux |

**Comment 10**

*Table 2: Is DSL the same as D in Equation 3? Make sure the notation is consistent.*

**Response 10**

180 $D_{SL}$ from table 2 has been changed to D.

**Comment 11**

*Equation 8: Should the Pimm term be negative in this equation? Immobilized P would be subtracted from the inorganic P pool. Also, shouldn't equation 9 by included as a negative term in Equation 8 since*

 *sorption reduces the inorganic P pool? These equations don't seem to be mass balanced with respect to each other.*

**Response 11**

Thank you for noticing. Yes, it is a typo. $P_{imm}$ should be negative. Regarding equation 9, we understand why it may seem that way. The equation follows Goll et al. (2017), that at the same time follows a previous study by the same author (Goll et al. , 2012). In Goll et al. (2012) things get more clearer, the sum of $P_{sorb}$ and $P_{soil}$ represents the inorganic P pool. Both pools are assumed to be in equilibrium in daily basis. The only loss comes from the strongly sorb pool represented in the equation by $T_{sorb}$. Hence, the inorganic pool is first calculated as the fraction sorbed and the fraction that was not sorbed (given a map).The sum of the two lose phosphorus in a rate defined by the multiplication of $T_{sorb}$ by $P_{sorb}$. Then both get recalculated based on the fraction given by the map.

$$\frac{dPsoil}{dt} = (1 - K_s)(P_{wea} + P_{litmin} + P_{orgmin} - P_{leach} - P_{up} - \tau_{sorb}P_{sorb} - P_{imm}), \tag{9}$$

The estimation of $Psoil$ based on Goll et al. (2017), is originally taken from Goll et al. (2012). Here, the sum $P_{sorb}$ and $P_{soil}$ constitute the inorganic P pool in soil. Hence, the loss given by the rate of strong sorption is applied to the total inorganic P pool.

**Comment 12**

*Line 186: Is QD here the same as q in equation 6? Both are described as runoff.*

**Response 12**

They are both the runoff from different perspectives. $Q_D$ represents the runoff per soil layer, while q represents the total runoff from the soil layers.

**Comment 13**

205 Line 189-190: Similar to C:N, the variable C:P ratio of leaf and root tissues is mentioned here but there is no explanation or equation for what controls the value of the ratio.

**Response 13**

We have added the following text to section 2.3.3, to introduce and explain the equation:

$$CP_{leaf} = \frac{C_{leaf}}{P_{leaf}}, \tag{10}$$

210 where $C_{leaf}$ is the carbon content in leafs and $P_{leaf}$ is the phosphorus content in leafs. $CP_{leaf}$ is one of the most important nutrient limitators in the model. The limiting effect of $CP_{leaf}$ is when its value is lower than the maximum CP leaf ratio parameter $CP_{leafmax}$. This leads to biomass reduction. In contrast to $CN_{leaf}$, $CP_{leaf}$ does not control the maximum carboxilation rate of RuBISCO. A more detailed description of nutrient limitation can be found in section 2.4.

215 **Comment 14**

*Line 195: the vegetation P change over time*

**Comment 14**

Thanks for noticing. You are certainly correct. It has been changed to:

220 where $V_{egp}$ is the vegetation P change over time,

**Comment 15**

*Line 200-204: Equation 16 needs some conceptual explanation. It's not directly clear from the equation and description what process this is representing. Are the nitrogen costs related to actual nitrogen availability?*

**Response 15**

The following has been added to the description of equation 16:

Here, the nitrogen cost refers to the nitrogen required for protein structures involved in the metaboliza-
tion of phosphorus in plants.

**Comment 16**

*Line 232: I did not find an explanation of CPleafmax, CNleafmax, Rleafp, or RleafN in the text or
equations showing how the model depends on these parameters. If these parameters are important enough
to be the basis for the sensitivity analysis, they should be clearly explained in the text.*

**Response 16**

Than you for your comment. The following text has been added:

$CP_{leafmax}$ and $CN_{leafmax}$ are the maximum leaf CP or CN ratios respectively. If the values of $CP_{leaf}$ and
$CN_{leaf}$ are above these thresholds the model will take the system to be nutrient limited by either P or N.
$R_{leafN}$ and $R_{leafP}$ are parameters that represents the resorption of nitrogen and phosphorus in leafs. This
 partly controls the loss of nitrogen and phosphorus from vegetation to the litter pool. $V_{maxp}$ and $V_{maxn}$
are the P and N maximum uptake rates.

**Comment 17**

*Line 259-261: This is not shown in any of the figures. This statement could be supported by showing a
map of biomass from the different simulations and the difference from the C-only simulation*

**Response 17**

Thank you for your comment. We do show this later in the paper. Figure 4 shows the difference in PFTs fractions. Although not directly, broadleaf trees PFTs fractions are shown to be reduced in CNP compared to C only. This reduction indicates a loss of wood biomass and supports the statement in Line 259-261.

The high GPP in the baseline simulation can be explained by the overestimation of the vegetation biomass especially broadleaf trees in tropical regions as stated in Mengis et al. (2020). The representation of vegetation biomass is linked to the PFTs fractions in the model.

**Comment 18**

*Line 274: Difference in tropical vegetation biomass is also not shown in any figure. This could be shown as a map or an average biomass value by latitude for different simulations.*

**Response 18**

Thank you for your comment. We do show this later in the paper. Figure 4 shows the difference in PFTs fractions. Although not directly, broadleaf trees PFTs fractions are showed to be reduced in CNP compared to C only. This reduction indicates a loss of wood biomass and supports the statement in Line 274.

While CNP (72 Pg C yr-1 ) resulted in lower values, due to the reduction of tropical vegetation biomass. The reduction of tropical biomass mainly in broadleef trees carbon is reflected in the fraction of the PFT shown in the model output.

**Comment 19**

*Line 294: I would reorder the figures so they appear in the order described – 6 is described before 4 and 5.*

**Response 19**

The figures were reordered as suggested.

**Comment 20**

270   *Line 304-306: This statement should be supported by a figure showing vegetation carbon as a map or latitudinal gradient.*

**Response 20**

Thank you for your comment. Figure 4 shows the difference in PFTs fractions. Although not directly, broadleaf trees PFTs fractions are showed to be reduced in CNP compared to C only. This reduction

275   indicates a loss of wood biomass and supports the statement in Line 304-306.

**Comment 21**

*Line 314: Figure 7 does not show the difference in N compared to Wania et al. (2012), so this statement cannot be evaluated.*

**Response 21**

280   You are correct, there is not direct comparison in the paper in regards with this statement. The only way is to search for Wania et al. (2012) paper. The paragraph was change to:

Globally, the CNP simulated soil nitrogen is reduced compared to the original N structure in the UVic ESCM version 2.9 presented by Wania et al. (2012). The primary differences between Wania et al. (2012)

285   N cycle and the current version are the soil layer structure and the stochiometry response to N limitation. In the former, N could be transfer from other pools when N was outside of the ratios threshold and thereby be considered to be limiting vegetation.

**Comment 22**

*Figure 4: This figure was difficult to interpret because only the differences in PFT fractions were shown, and not the actual fractions. There also is not much explanation of how relative PFT distributions relate to N and P cycling in the model so it's not clear how relevant this is to the main model developments being described.*

**Response 22**

The answer to the comment is in line with comment 3, 13 and the limitation description of the model. The biomass reduction effect of N and P affect PFTs in different ways based on $CN_{leafmax}$ and $CP_{leafmax}$ parameters. Understanding the effect of P and N is quite insightful as we can observed how different type of vegetation are responding to the limitation. This allows for a better representation of vegetation biomass and shows the degree of biomass reduction that our parameters and constrained are doing. From a different perspective, looking and N and P as a tool to represent the terrestrial system as closer to nature as possible N and P effect to PFTs gains a lot of value.

**Comment 23**

*Figure 5: There is no explanation of how these correlations are calculated. Is this based on relative amount of all PFTs in each grid cell? This does not seem to be the most useful test of the model since many grid cells are dominated by one or two PFTs. Wouldn't variation in PFTs across grid cells be a more useful metric to test?*

**Response 23**

The correlation encompasses PFTs across grid cells. We have adjusted the caption of Figure 5 to make this clear:

Figure 5. PFTs fractions across grid cells in the UVic ESCM for 2008-2012, CNP correlation to Poulter et al. (2015) PFTs dataset.

**Comment 24**

*Lines 326-328: There is an order of magnitude range in the different estimates, so they don't seem like a very strong constraint on the model. Is there any expectation of which set of estimates might be more accurate?*

**Response 24**

Yes, the difference between the lower range and higher range are relatively large. We can not state which estimate is more accurate, global nitrogen and phosphorus fluxes are currently largely uncertain. The likely answer is that the most accurate value holds the middle position among the range of models. This uncertainty becomes even larger for phosphorus.

**Comment 25**

*Line 329: Is CN ratio referring to soil, vegetation, or whole ecosystem? Figure 7 is also unclear about this.*

**Response 25**

It is referring to vegetation CN ratio. The title of the plot has been changed from "CN ratio" to "Vegetation CN ratio".

**Comment 26**

*Line 336: Equation 16 included some nitrogen cost of phosphatase parameters. Does this not connect the N and P cycles in a way that could allow co-limitation? It's hard to tell without more explanation of that equation.*

**Response 26**

You are technically correct. It was poorly written and has been be reworded in the text. In the model structure there are some co-limitation factors. We do not account directly for co-limitation. A description of the limitation in the model which has been added in section 2.4.

335

The model assumes nutrient limitation when the concentration of nitrogen or phosphorus is higher than the maximum CN ($CN_{leafmax}$) and CP ($CP_{leafmax}$) in leafs. For grids with nutrient limitation the carbon in leaf is reduced to match the maximum CN or CP ratios in leafs. The carbon that is removed is transferred to the litter pool. This reduction can happen for one or both nutrients until the ratio is met. The following
340 equations regulate the reduction of biomass based on nutrient limitation:

$$C_{leaflimitedn} = N_{leaf}CN_{leafmax}, \tag{11}$$

$$C_{leafdiffn} = C_{leaf} - C_{leaflimitedn}, \tag{12}$$

345 $$C_{leaflimitedp} = N_{leaf}CP_{leafmax}, \tag{13}$$

$$C_{leafdiffp} = C_{leaf} - C_{leaflimitedp}, \tag{14}$$

were $C_{leaflimitedn}$ and $C_{leaflimitedp}$ are the carbon concentration in leafs if the system is considered to be limited. $C_{leafdiffn}$ and $C_{leafdiffp}$ are the the carbon lost due to nutrient limitation and their value are sum
350 in the litterfall equation when the system is in nutrient limitation.

**Comment 27**

*Line 341: The model does not include anthropogenic N inputs, so is it reasonable to compare it with estimates that do include anthropogenic inputs? Couldn't this indicate that the model overestimates natural sources since anthropogenic N inputs in reality are very high?*

**Response 27**

We do account for atmospheric anthropogenic N deposition. We do not account for agricultural inputs in this paper. The model structure for $N_2O$ might indeed overestimate natural sources as you mentioned but are within the range of uncertainty. For a follow-up project (part of a paper in preparation) we applied agricultural fertilization and retested the model $N_2O$ and the resulting emission is increased from 10 to 25 %. The model estimation of anoxic fractions from wetlands might be mitigating the effect of agricultural fertilization. Nonetheless, the model does acceptably well.

**Comment 28**

*Line 372-374: Global terrestrial P should be included as a line in Table 6. Table 6 does not indicate estimates from terrestrial P models (or at least does not indicate which estimates are from models versus measurement syntheses). What is the evidence that other models are underestimating P in subsoils and not that this model overestimates P in subsoils?*

**Response 28**

Thank you for the comment. Global terrestrial P has been included in table 6. Table 6 was change accordingly. The statement is based only on the comparison with He et al. (2021) datasets. Hence, it was removed from the text.

**Comment 29**

*Figure 10: This figure doesn't make much sense to me. Why would the model N:P leaf ratio be perfectly linear with respect to latitude? Is the Reich and Oleksyn relationship a simple linear function with respect to latitude? If so, this seems like a very simplistic test of a complex model. Also, it is difficult to interpret this figure because there was no explanation provided for what controls leaf N:P ratio in the model.*

**Response 29**

It is a linear function with respect to latitude. This comparison was set to use the same metric as in Wang et al. 2010. While simplistic this shows if our model correctly displays the global limitation pattern where N is abundant in tropics and scarse in high latitudes and opposite to P. It is of course not that simplistic in nature and in reality without a linear function our model nor the observations describe a perfect line.

**Comment 30**

*Table A1: There is no reason this short table should be in a separate appendix. It's an important part of the model and should be in the main text.*

**Response 30**

Table A1 was moved to the main text after table 2 "Table 2. Updated nitrogen cycle parameters. See appendix A.1 for values that vary for each PFT."

**Comments from reviewer 2**

**General comment**

*Sisto et al. describe the modifications to the UVic ESCM intermediate complexity Earth system model done to represent new nitrogen (N) and phosphorus (P) cycling processes. The new model is evaluated based on global datasets of C, N, and P pools and fluxes. Incorporating P cycles into ecosystem models is timely and important work. The equations and processes are clearly described. Overall, this manuscript is well-organized and easy to read. However, some concerns need to be addressed or clarified.*

**Response**

Thank you for the positive feedback

**General comment**

*Insufficient description of methodology: The model is insufficiently described in 2.4 Nitrogen and phosphorus limitation. The critical aspect of nutrient effects on C cycling is the competition of plants and microbes for limited nutrients. This aspect is not described here. How do you deal with cases in which available soil P and mineralization are insufficient to satisfy the P immobilization demand? In addition, the N, P limitation on the C cycle in vegetation is described too simply; we don't have enough details. (The Parameters for sensitivity analysis are not described enough in the model structure).*

**Response**

Thank you for your comment. A more in depth description was added in section 2.4. You are correct the competition of plants and microbes for limited nutrients is indeed important for nutrient limitation. Having said that, the UVic ESCM is a global model and our vegetation is represented by a top-down approach. We do account for plant competition in the model and it is shown in the PFTs change in the text and figures. We do not account directly for microbial interaction. That by itself could be a great addition for our model but remains an avenue for future model development. Some aspect of nutrient cycling need

to be simplified. Hence, some interactions and dynamics are not captured within are global model structure. It is true that the parameters of the sensitivity analysis are not described sufficiently. A more detailed description was added.

Section 2.4:

The model assumes nutrient limitation when the concentration of nitrogen and phosphorus is higher than the maximum CN ($CN_{leafmax}$) and CP ($CP_{leafmax}$) in leafs. For grids with nutrient limitation the carbon in leaf is reduced to match the maximum CN or CP ratios in leafs. The carbon that is reduced is transferred to the litter pool. This reduction can happen for one or both nutrients until the ratio is met. The following equations regulate the reduction of biomass based on nutrient limitation:

$$C_{leaflimitedn} = N_{leaf}CN_{leafmax}, \tag{15}$$

$$C_{leafdiffn} = C_{leaf} - C_{leaflimitedn}, \tag{16}$$

$$C_{leaflimitedp} = N_{leaf}CP_{leafmax}, \tag{17}$$

$$C_{leafdiffp} = C_{leaf} - C_{leaflimitedp}, \tag{18}$$

where $C_{leaflimitedn}$ and $C_{leaflimitedp}$ are the carbon concentration in leafs if the system is considered to be limited. $C_{leafdiffn}$ and $C_{leafdiffp}$ are the the carbon lost due to nutrient limitation and their value are sum in the litterfall equation when the system is in nutrient limitation.

Parameter sensitivity:

$CP_{leafmax}$ and $CN_{leafmax}$ are the maximum leaf CP or CN ratios respectively. If the values of $CP_{leaf}$ and $CN_{leaf}$ are above these thresholds the model will take the system to be nutrient limited by either P or N.

$R_{leafN}$ and $R_{leafP}$ are parameters that represents the resorption of nitrogen and phosphorus in leafs. This partly controls the loss of nitrogen and phosphorus from vegetation to the litter pool. $V_{maxp}$ and $V_{maxn}$ are the P and N maximum uptake rates.

**General comment**

440 *Lack of evaluation: Even though this is an ESM model, this paper focuses more on land surface processes. The evaluation should include different scales, such as site and regional levels. We can only see the global scale values. In addition, do you capture spatial gradients in GPP or seasonal or interannual variation in GPP?*

**Response**

445 Thank you for your comment. While it is true that a site or regional level comparison would be beneficial for the paper, the UVic ESCM resolution is not suitable for those type of comparison. Depending on the definition of what is a region we might present results of nutrient variables. However, the lack of observations in a region large enough to be compatible for comparison is a challenge. There are examples of studies that compared nitrogen and phosphorus in sites (e.g. Goll et al. (2017); Nakhavali et al. (2021)).

450 In those studies the models have considerably higher resolution that ours and the land-surface can be de-coupled from the atmosphere, allowing prescribed local vegetation and meteorology . Hence, a site comparison is more feasible. Furthermore, the UVic ESCM is always used for global studies. Thereby, a global approach was thought to be the most reasonable. We added the following figure representing seasonal and global GPP compared to FLUXCOM GPP dataset.

[Figure]

**Figure 2.** Upper left figure shows FLUXCOM GPP dataset from 2000-2010, upper right figure shows the seasonal GPP from 1990-2015 for Baseline, CN and CNP. The second line from left to right shows the global GPP from 2000-2010 for Baseline, CN and CNP. The third line from left to right shows the difference between Baseline, CN and CNP and FLUXCOM GPP dataset. The fourth line from left to right shows the correlation of Baseline, CN and CNP to FLUXCOM GPP dataset.

455     The GPP distribution from Baseline, CN and CNP reproduce FLUXCOM dataset values reasonably well. The seasonal pattern of GPP is also well represented within out simulations as shown in Fig. 3. The addition of nutrients improves the representation of GPP, where CNP had the highest correlation with FLUXCOM GPP dataset.

**General comment**

460   *I expect the author to provide an understanding of the model dynamics, such as the C, N, and P processes in vegetation and soil, and how they couple and interact.*

**Response**

Thank you for your comment. A more detailed description has been added in section 2.4 describing the coupling of nutrient limitation.

465

    The model assumes nutrient limitation when the concentration of nitrogen or phosphorus is higher than the maximum CN ($CN_{leafmax}$) and CP ($CP_{leafmax}$) in leafs. For grids with nutrient limitation the carbon in leaf is reduced to match the maximum CN or CP ratios in leafs. The carbon that is reduced is removed to the litter pool. This reduction can happen for one or both nutrients until the ratio is met. The following

470 equations regulate the reduction of biomass based on nutrient limitation:

$$C_{leaflimitedn} = N_{leaf}CN_{leafmax}, \tag{19}$$

$$C_{leafdiffn} = C_{leaf} - C_{leaflimitedn}, \tag{20}$$

475  $$C_{leaflimitedp} = N_{leaf}CP_{leafmax}, \tag{21}$$

$$C_{leafdiffp} = C_{leaf} - C_{leaflimitedp}, \tag{22}$$

were $C_{leaflimitedn}$ and $C_{leaflimitedp}$ are the carbon concentration in leafs if the system is considered to be limited. $C_{leafdiffn}$ and $C_{leafdiffp}$ are the the carbon lost due to nutrient limitation and their value are sum in the litterfall equation when the system is in nutrient limitation.

**Specific comments**

**Comment 1**

*Line 68, 71: Some references related to the recent CNP model should include. (Fleischer et al., 2019; Thum et al., 2019; Wang et al., 2020; Yang et al., 2014)*

**Response 1**

Thank you for the comment. The suggested references were included in the text.

of P into ESM structures is possible and that it improves the representation of vegetation biomass in tropical regions (Wang et al. , 2007, 2010; Goll et al. , 2012, 2017; Fleischer et al. , 2019; Thum et al. , 2019; Yang et al. , 2019; Wang et al. , 2020; Nakhavali et al. , 2021).

**Comment 2**

*Line 153: It is section 2.4, not 2.5*

**Response 2**

Thank you noticing. The line was changed accordingly.

Section 2.4 presents a detailed explanation of nutrient limitation for N and P.

**Comment 3**

*Line 155: The names of inorganic P pools need to be more consistent. You named the labile P pool, but in figure 1, it is Dissolved Inorganic P. Different models have a different definitions of labile P and dissolved inorganic P, such as (Goll et al., 2017; Thum et al., 2019; Wang et al., 2020; Yang et al., 2014). So, giving clear definitions and keeping the name consistent is better.*

**Response 3**

Thank you noticing. Dissolved inorganic P was changed to labile P thorough the text accordingly.

**Comment 4**

*Line 170: Which part of the approach is more controllable? Explain it more precisely.*

**Response 4**

Thank you for your comment. An explanation has been added to the line.

Here we only apply Wang et al. (2010) approach as we found it to be more controllable and an advantage to the planned coupling of P flux from land into the ocean. Hartmann et al. (2014) requires the estimation of runoff by the model structure. Hence, while representing a dynamical P release it needs to be carefully assessed so that no extreme overestimation or underestimation are represented regionally. Wang et al. (2010) approach provides constant input without variability which in this particular case is favorable.

**Comment 5**

*Line 221: In equation 19, if the CN increases, the Vcmax will increase; how is plant productivity reduced? Line 219 describes it in the opposite direction.*

**Response 5**

Thank you for your comment. In equation 9 we multiply $CN_{leafi}$ not $CN_{leaf}$. $CN_{leafi}$ is described as the inverse of $CN_{leaf}$. Hence, line 219. $CN_{leafi}$ was changed to $CN_{invleaf}$.

**Comment 6**

*Line 224: The NP limitation needs to be clarified. Is it based on Liebig's Law of the Minimum?*

**Response 6**

Thank you for your comment. No, the nutrient limitation has a more simplistic approach. The following was added in section 2.4.

The model assumes nutrient limitation when the concentration of nitrogen and phosphorus is higher than the maximum CN ($CN_{leafmax}$) and CP ($CP_{leafmax}$) in leafs. For grids with nutrient limitation the carbon in leaf is reduced to match the maximum CN or CP ratios in leafs. The carbon that is reduced is transferred to the litter pool. This reduction can happen for one or both nutrients until the ratio is met. The following equations regulate the reduction of biomass based on nutrient limitation:

$$C_{leaflimitedn} = N_{leaf} CN_{leafmax}, \tag{23}$$

$$C_{leafdiffn} = C_{leaf} - C_{leaflimitedn}, \tag{24}$$

$$C_{leaflimitedp} = N_{leaf} CP_{leafmax}, \tag{25}$$

$$C_{leafdiffp} = C_{leaf} - C_{leaflimitedp}, \tag{26}$$

were $C_{leaflimitedn}$ and $C_{leaflimitedp}$ are the carbon concentration in leafs if the system is considered to be limited. $C_{leafdiffn}$ and $C_{leafdiffp}$ are the the carbon lost due to nutrient limitation and their value are sum in the litterfall equation when the system is in nutrient limitation.

**Comment 7**

*Line 248: Phosphorus dataset is Pdataset in the equation (20).*

**Response 7**

Thank you for noticing. The lines has been changed to:

the soil layer depth and $P_{dataset}$ (kg P (kg soil)-1) is He et al. (2021) dataset.

**Comment 8**

*Line 273: Should the baseline have a higher NPP? If I understand it correctly.*

**Response 8**

Only for CN not CNP. This was observed since the first construction of the nitrogen cycle in the UVic ESCM: Wania et al. (2012), argued that the reason behind the high NPP was the dependence of autotrophic respiration on N content in leaf, root and stem which are based on the original MOSES/TRIFFID version (Cox et al. , 1998). CNP does result in a lower value.

**Comment 9**

*Line 277: This part is hard to follow and needs to be articulate.*

**Response 9**

Thank you for your comment. The paragraph was rewritten to:

In CN and CNP, the reduction of wood CN ratios and higher leaf content than in CN and CNP which
fluctuates from a minimum to a maximum value gives place to the reduction of the maintenance respi-
ration which reduces the autotrophic respiration and consequently NPP. Furthermore, in the new CNP
version while wood CN remains to be fixed the stochiometrical reduction of wood carbon by the lack of P
availability decreases wood carbon even more especially in tropical forests and other tropical ecosystems.

**Comment 9**

*Line 288: In fig 3, why CN has a larger atmospheric CO2 pool than CNP?*

**Response 9**

The main driver for that response is the increase of ocean carbon uptake, given the initial reduction of
land carbon sink in CNP. That offsets the effects of phosphorus. It gets a bit more complicated than this.
Given our model limiting structure in this study phosphorus limitation had low impact. We recently have
assessed that atmospheric $CO_2$ is indeed lower in CNP for 2020 under a higher impact of phosphorus
limitation in the system. The impact is determined by $CP_{leafmax}$, the maximum phosphorus in leafs set in
the model.

**Comment 10**

*Line 298: Could you describe your mechanisms of dynamic PFT, maybe in the supplement?*

**Response 10**

The following was added to the description of TRIFFID in line 87:

The terrestrial vegetation is simulated by a top-down representation of interactive foliage and flora including dynamics (TRIFFID) representing vegetation interaction between 5 functional plant types: broadleaf trees, needleleaf trees, shrubs, C3 grasses, and C4 grasses that compete for space in the grid following the Lotka-Volterra equations (Cox , 2001). Net carbon fluxes estimated in the model updates the total areal coverage, leaf area inxes and canopy height for each PFT. For reach PFT the carbon fluxes are derived from a photosyntesis-stomatal conductance model (Cox et al. , 1998).

**Comment 11**

*Line 329: Tropical areas are supposed to have enough N, so a lower NP ratio.*

**Response 11**

We are unsure of exactly what the reviewer was asking in this comment. Line 329: The global pattern of CN ratio is similar to Wania et al. (2012) structure with the highest located in tropical regions especially. Hence, it's not NP but CN.

**Comment 12**

*Line 332: uncertainty (95-730). This is a huge range.*

**Response 12**

Thank you for your comment. Yes, depending of the species the C:N ratio can vary largely. For example, Martin et al. (2014) shows a range of tropical wood C:N ratio from 92 to 1360 in different Panamanian tree species.

**Comment 13**

*Line 354: I wondered why Fertilization inputs did not use here.*

**Response 13**

This paper represents the core skeleton of the phosphorus cycle. The phosphorus inputs data and structure were not at the point of utilization. Hence, we decided to only account for a natural P cycle. This is a priority in future development plans. Furthermore, P fertilization forcing is not yet available in future scenarios.

**Comment 14**

*Line 366: "This underestimation is likely the result of a high mineralization rate." Do you check the soil C pool? And maybe the CP ratio also gives some clues about the underestimation of organic P.*

**Response 14**

After writing the paper and analyzing further simulations. We've realised that the underestimation of P comes simply by the lack of P fertilizers. Hence line 366 will be changes to: This underestimation is likely the result of the lack of P fertilization on land.

 **Comments from reviewer 3**
* * *
**General comment**

*The paper is mostly presenting a new model, which is appropriate for this journal. However, more "science" needs to be presented with the model in order to determine if their new tool is appropriate for further studies at its present state. For example, the authors could provide more model validation and intercomparison with available products and similar models. I appreciate the evaluation of the Keeling curve and FLUXCOM GPP, but this is only a validation of CO2. What about energy? What about water? What about the nutrient cycles?*

**Response**

Thank you for your comment. The purpose of this paper was to describe a terrestrial nitrogen and phosphorus cycle adapted, developed and implemented for the UVic ESCM version 2.10. As such the main changes we wanted to capture in this paper are in the terrestrial system, especially the vegetation. While the Keeling curve does only validate $CO_2$, the FLUXCOM GPP validates the gross primary productivity, a relevant variable to assess the representation of the metabolism in vegetation. While we appreciate your suggestions, assessing energy and water were recently extensively evaluated for the latest public version of the model (UVic ESCM 2.10) by Mengis et al. (2020). The addition of terrestrial N and P cycles had only a minor effect of these variables, and thus reassessing them would add considerable length and little value to the present paper. Finally, we do validate nutrient cycles with other relevant modelling studies and available observations, and through our revisions have made this clearer. One of the challenges of representing nutrients in Earth system models is the lack of observations from where to validate.

**General comment**

*I suggest adding tool like ILAMB (Collier et al., 2018) for evaluating model performance throughout variables related to the carbon, water and energy cycles, as well as a sensitivity analysis of current with meteorological variables.*

**Response**

635  Thank you for your comment. A detailed evaluation was conducted recently for the UVic ESCM version 2.10 by Mengis et al. (2020). Regarding the carbon cycles, we did use ILAMB dataset for GPP (FLUX-COM). As for the nutrient cycles, ILAMB is not yet adequate to assess nitrogen or phosphorus in ESMs (Spafford and MacDougall , 2021).

The following text has been added in section 2.1:

640

Mengis et al. (2020) merged previous version of the UVic ESCM and evaluated its performance representing carbon and heat fluxes, water cycle and ocean tracers.

**General comment**

*The authors should add comparisons with other similar studies. They do that to an extent (Poulter et al.,*
645  *2015; He et al., 2021), but there are so many other similar and recent studies that should be added, such as: Wang, Y., Ciais, P., Goll, D., Huang, Y., Luo, Y., Wang, Y.P., Bloom, A.A., Broquet, G., Hartmann, J., Peng, S., Penuelas, J., Piao, S., Sardans, J., Stocker, B.D., Wang, R., Zaehle, S., Zechmeister-Boltenstern, S., 2018. GOLUM-CNP v1.0: A data-driven modeling of carbon, nitrogen and phosphorus cycles in major terrestrial biomes. Geosci. Model Dev. 11, 3903–3928. https://doi.org/10.5194/gmd-11-3903-2018. Goll,*
650  *D.S., Vuichard, N., Maignan, F., Jornet-Puig, A., Sardans, J., Violette, A., Peng, S., Sun, Y., Kvakic, M., Guimberteau, M., Guenet, B., Zaehle, S., Penuelas, J., Janssens, I., Ciais, P., 2017. A representation of the phosphorus cycle for ORCHIDEE (revision 4520). Geosci. Model Dev. https://doi.org/10.5194/gmd-10-3745-2017. Braghiere, R.K., Fisher, J.B., Allen, K., Brzostek, E., Shi, M., Yang, X., Ricciuto, D.M., Fisher, R.A., Zhu, Q., Phillips, R.P., 2022. Modeling global carbon costs of plant nitrogen and phosphorus*
655  *acquisition. J. Adv. Model. Earth Syst. e2022MS003204. https://doi.org/10.1029/2022MS003204*

**Response**

Thank for your comment. In our study we compare our results with several other studies such as: Reich and Oleksyn (2004); Mengis et al. (2020); Wania et al. (2012); Global Soil Data Task Group (2000);

Zaehle et al. (2010); Yang et al. (2009); Xu-ri and Prentice (2008); Crippa et al. (2021); Smil (2000); Mackenzie (2002); Wang et al. (2010); Yang et al. (2013).

We thank you for the suggested sources. Many of these papers use experimental designs examining much smaller spatial scales and hence are difficult to compare with results from our experimental design. For example, while some equations we use are based on Goll et al. (2017), in their study a regional approach is used, which cannot be compared to our global study. There are indeed other studies that address phosphorus in regional scales, but such studies cannot easily be scaled-up to the global scale, and nor can global models be scaled-down to the site-scale for comparison.

The following comparison have been added:

Section 3.2.1:

Our model estimates a mean biological nitrogen fixation for 2010-2020 of 119 Tg N yr$^{-1}$. This value is above 35 Tg N yr$^{-1}$ from Braghiere et al. (2022) and within the range of 52–130 Tg N yr$^{-1}$ presented by Barnard and Friedlingstein (2020)

Section 3.2.2:

The total amount of vegetation nitrogen (2.20 Pg N) was lower than the previous N cycle (2.94 Pg N, Wania et al. (2012)). These values are similar to Zaehle et al. (2010) (3.8 Pg N) and Wang et al. (2018) (3.9 PgN) but lower than Li et al. (2000) (16Pg N) and Yang et al. (2009). (18 Pg N). Our tropical (30 to 45gN m-2 ) and boreal forest vegetation nitrogen (20 to 35gN m-2 ) results are lower than from Wania et al. (2012) (30 to 40gN m-2 ), and those of Xu-ri and Prentice (2008) and Yang et al. (2009) (both studies ranged between of 150 to 400 gN m-2 ).

Table 6:

| Variables | Value (Pg P) | References (Pg P) |
|---|---|---|
| Total inorganic P | 20.8 | 35-40 Smil (2000) |
| | | 36 Mackenzie (2002) |
| | | 26.5 Wang et al. (2010) |
| | | 13.7 Wang et al. (2018) |
| Total organic P | 3.5 | 5-10 Smil (2000) |
| | | 5 Mackenzie (2002) |
| | | 5.7 Wang et al. (2010) |
| | | 8.6 Yang et al. (2013) |
| Labile P | 1.4 | 1.5 Wang et al. (2010) |
| | | 3.6 Yang et al. (2013) |
| Sorbed P | 1.1 | 1.7 Wang et al. (2010) |
| Strongly sorbed P | 12 | 7.6 Wang et al. (2010) |
| Occluded | 6.3 | 9.0 Wang et al. (2010) |
| Vegetation P | 0.2 | 0.4 Wang et al. (2010) |
| | | 0.5 Smil (2000) |
| | | 0.5 Wang et al. (2018) |
| | | 0.2 Wang et al. (2018) |
| | | 0.5 Mackenzie (2002) |
| P Litter | 0.01 | 0.04 Wang et al. (2010) |
| | | 0.03 Wang et al. (2018) |

**General comment**

*Moreover, why is this model needed? Is it just another model on top of the CMIP simulations? A deeper discussion about how this model relates to other models and the future of climate modeling is needed.*

**Response**

Thank you for your comment. The following text has been added to the discussion:

The UVic ESCM has been a critical tool in developing the cumulative emissions framework to climate mitigation (Zickfeld et al. , 2009; Matthews et al. , 2009; Matthews and Weaver , 2009; MacDougall and Knutti , 2016; Mengis et al. , 2018; Tokarska et al. , 2019) due to its low computations cost and strict enforcement of matter and energy conservation the model is capable of conducting a host of simulation beyond the limits of most other models, but with higher resolution than other EMICs (e.g. Montenegro et al. (2007); Matthews and Caldeira (2008); Keller et al. (2014); MacDougall and Knutti (2016); MacDougall (2017); Pahlow et al. (2020); Kvale et al. (2021)) . As terrestrial nutrient limitation constrains the carbon cycle in nature, the new nitrogen and phosphorus modules allows addressing research questions relating carbon budgets, carbon cycle and $CH_4$ feedbacks, carbon dioxide removal and permafrost carbon cycle, among other questions. Furthermore, the nitrogen and phosphorus cycles can represent environmental and climate critical processes such as the release of $N_2O$, agricultural impacts on terrestrial soils and coastal lines, eutrophication, anoxic events and nutrient fluxes from land to ocean.

**General comment**

*Why do you have a schematic of the P cycle but not the N cycle? Be consistent.*

**Response**

Thank you for the suggestion. The following figure has been added:

[Figure]

**Figure 3.** Diagram representing the UVic ESCM nitrogen cycle.

**Specific comments**

**Comment 1**

*Introduction needs work in properly linking biodiversity with other aspects of the biogeochemical cycles in the Earth system and climate change. I added a few extra references, but a more thorough literature review is required. Introduction: I find the introduction a bit shallow and very model centric. I understand this is a modeling journal, but the reader would benefit from more general scientific discussions at the beginning. You may want to cite: Wieder, W.R., Cleveland, C.C., Smith, W.K., Todd-Brown, K., 2015. Future productivity and carbon storage limited by terrestrial nutrient availability. Nat. Geosci. 8, 441–444. https://doi.org/10.1038/ngeo2413 Zaehle, S., Jones, C.D., Houlton, B., Lamarque, J.-F., Robertson, E., 2015. Nitrogen Availability Reduces CMIP5 Projections of Twenty-First-Century Land Carbon Uptake. J. Clim. 28, 2494–2511. https://doi.org/10.1175/JCLI-D-13-00776.1*

**Response**

Thank you for the comment. The following text was added to the introduction:

Biodiversity plays a crucial role in biogeochemical cycles. Microbial diversity for example enables nitrogen pathways that only some taxa can metabolize. Plant diversity, is linked to soil health and functioning, and is core for the nitrogen and phosphorus cycles. Overall, biodiversity constitute an environmental resilience factor to abrupt changes (Van Oijen et al. , 2020). However, implementing such dynamics remains far beyond the capabilities for the present generation Earth systems models.

**Comment 2**

*Line 2: Earth system models (ESMs)*

**Response 2**

Changed

**Comment 3**

*Line 6: Nitrogen (N). It is not the first time the word nitrogen appears. Please define acronyms in first appearance.*

**Response 3**

Thank you for your comment Nitrogen (N) was defined in line 4.

**Comment 4**

*Line 30: Missing References:*

 **Response 4**

Thank you for your comment. Goll et al. (2017) and Wang et al. (2020) were added as references.

**Comment 5**

*Line 63: This isn't true. Although ESM modeling with phosphorus is indeed limited. See:Wang, Y., Ciais, P., Goll, D., Huang, Y., Luo, Y., Wang, Y.P., Bloom, A.A., Broquet, G., Hartmann, J., Peng, S., Penuelas, J., Piao, S., Sardans, J., Stocker, B.D., Wang, R., Zaehle, S., Zechmeister-Boltenstern, S., 2018. GOLUM-CNP v1.0: A data-driven modeling of carbon, nitrogen and phosphorus cycles in major terrestrial biomes. Geosci. Model Dev. 11, 3903–3928. https://doi.org/10.5194/gmd-11-3903-2018. Goll, D.S., Vuichard, N., Maignan, F., Jornet-Puig, A., Sardans, J., Violette, A., Peng, S., Sun, Y., Kvakic, M., Guimberteau, M., Guenet, B., Zaehle, S., Penuelas, J., Janssens, I., Ciais, P., 2017. A representation of the phosphorus cycle for ORCHIDEE (revision 4520). Geosci. Model Dev. https://doi.org/10.5194/gmd-10-3745-2017. Braghiere, R.K., Fisher, J.B., Allen, K., Brzostek, E., Shi, M., Yang, X., Ricciuto, D.M., Fisher, R.A., Zhu, Q., Phillips, R.P., 2022. Modeling global carbon costs of plant nitrogen and phosphorus acquisition. J. Adv. Model. Earth Syst. e2022MS003204. https://doi.org/10.1029/2022MS003204*

**Response 5**

Neglected was changed for rare.

**Comment 6**

*The Baseline GPP of the model substantially overestimates FLUXCOM, why is that?*

**Response 6**

Thank you for your comment. It is explained in the paper in section 3.1.1 line 258: The high GPP in the baseline simulation can be explained by the overestimation of the vegetation biomass especially broadleaf trees in tropical regions stated in Mengis et al. (2020).

**Comment 7**

*Line 274: NPP of 72 PgCyr also seems a bit high. What is a good estimate of global NPP? Add values in the study.*

**Response 7**

That is correct, we are very close to the upper range of the modelled NPP range Li et al. (2015): 21.5 to 69.3 Pg C yr$^{-1}$. The following was added in the line:

Similar to Wania et al. (2012), we found higher values of NPP for CN (77.4 Pg C yr$^{-1}$) compared to the baseline simulation (74.2 Pg C yr$^{-1}$). While CNP (72 Pg C yr$^{-1}$) resulted in lower values, due to the reduction of tropical vegetation biomass. CN and CNP results are close to the upper range (21.5 to 69.3 Pg C yr$^{-1}$) of simulated NPP showed in Li et al. (2015).

**Comment 8**

*Figure 7. These values could be compared to other studies as well. See: Braghiere, R.K., Fisher, J.B., Allen, K., Brzostek, E., Shi, M., Yang, X., Ricciuto, D.M., Fisher, R.A., Zhu, Q., Phillips, R.P., 2022. Modeling global carbon costs of plant nitrogen and phosphorus acquisition. J. Adv. Model. Earth Syst. e2022MS003204. https://doi.org/10.1029/2022MS003204*

**Response 8**

Thank you for your comment. We are not sure what would be the basis for the comparison with the reference given. Braghiere et al. (2022) is focused on the plant cost for nutrient acquisition. Thereby, the study is centered on uptake. Figure 7 shows the vegetation CN ratio, vegetation nitrogen and soil nitrogen.

**Comment 9**

*Section 4: Limitations and applications of the terrestrial nutrient modules. I would add the role of mycor-*
*rhizae into NP acquisition. Please refer to: Braghiere, R.K., Fisher, J.B., Fisher, R.A., Shi, M., Steidinger,*
*B.S., Sulman, B.N., Soudzilovskaia, N.A., Yang, X., Liang, J., Peay, K.G., Crowther, T.W., Phillips, R.P.,*
*2021. Mycorrhizal Distributions Impact Global Patterns of Carbon and Nutrient Cycling. Geophys. Res.*
*Lett. 48. https://doi.org/10.1029/2021GL094514. Shi, M., Fisher, J.B., Brzostek, E.R., Phillips, R.P., 2016.*
*Carbon cost of plant nitrogen acquisition: global carbon cycle impact from an improved plant nitrogen cy-*
*cle in the Community Land Model. Glob. Chang. Biol. 22, 1299–1314. https://doi.org/10.1111/gcb.13131*

**Response 9**

Thank you for the suggestion, the following was added in Section 4:

5. The model does not account for uptake constrains on terrestrial vegetation. This includes spatial representations of mycorrhizal associations and the carbon cost of nitrogen and phosphorus uptake from soil (Shi et al. , 2016; Braghiere et al. , 2021, 2022). Furtermore, we do not estimate nitrogen cost for phosphorus metabolization or viseversa.

**References**

[revised manuscript text omitted]

---

## Author Response (AR2)

**Response to reviewers**

We appreciate the time and effort that the editors have dedicated to providing your valuable feedback on our manuscript. The reviews are copied verbatim and are italicized. Author responses are in regular font. Changes made to the manuscript are blue.

**Comments from reviewer 1**

**Comment**

*Lines 14-15: This sentence needs rewording. I suggest "... nutrient limitation resulted in a reduction of GPP from the Carbon-only value of 143 PgC/yr to 133 PgC/yr in the CN version and 129 PgC/yr in the CNP version."*

**Response**

Thank you for your comment. As suggested line 14-15 was changed to:

For the years 2001-2015 the nutrient limitation resulted in a reduction of GPP from the Carbon-only value of 143 PgC yr$^{-1}$ to 130 PgC yr$^{-1}$ in the CN version and 127 PgC yr$^{-1}$ in the CNP version.

**Comment**

*Line 123: This should read "if leaf C:N ratio is higher than CNleafmax..." Nitrogen limitation occurs when nitrogen concentration is lower, not higher, because C:N ratio divides by nitrogen.*

**Response**

Thank you for your comment. Line 123 has been changed to:

If leaf C:N ratio is higher than $CN_{leafmax}$ (the maximum CN ratio parameter) terrestrial vegetation biomass is reduced.

**Comment**

*Figure 1: The litter box should be labeled "Litter N", not "Litter P."*

25 ## Response

Thank you for your comment. Figure 1 litter box has been relabeled to Litter N:

[Figure]

**Figure 1.** Diagram representing the UVic ESCM nitrogen cycle.

**Comment**

30 *Line 226: P limitation of leaf biomass should occur when C:P ratio is higher than the maximum (implying that there is not enough P or too much C), not lower than the maximum.*

**Response**

Thank you for your comment. You are correct. Line 226 has been changed to:

35 The limiting effect of $CP_{leaf}$ is when its value is higher than the maximum $CP_{leaf}$ ratio parameter $CP_{leafmax}$.

**Comment**

*Equation 20: Is $\lambda UPtase$ in the numerator supposed to be $\lambda Ptase$? I did not see any explanation of $\lambda UPtase$ in the description. Also, is this equation a constant value? All the terms appear to be constants*
40 *rather than variables but it wasn't clear from the text if Ptase is expected to vary or if it is a constant in the model.*

**Response**

Thank you for your comment. That is a typo, $\lambda_{UPtase}$ is $\lambda_{Ptase}$. In equation 20 $\lambda_{UPtase}$ has been changed to:

45 $\lambda_{Ptase}$

Futhermore, the following line has been added:

$P_{tase}$ is a constant value.

**Comment**

*Line 259: This is incorrect. The model assumes nutrient limitation when the C:N ratio or C:P ratio (not N or P concentration) is higher than the maximum ratio. A concentration can't be compared to a ratio.*

**Response**

Thank you for your comment. Line 259 has been changed to:

The model assumes nutrient limitation when the estimated CN and CP leaf ratio is higher than the maximum CN (CN$_{leafmax}$) and CP (CP$_{leafmax}$) ratio in leafs.

**Comment**

*Line 306: Figure 3 in the revised manuscript shows a value of about 130 for the baseline simulation, which*
 *is different from the text and also very different from the figure in the previous version of the manuscript which did have a value of about 143 as the text states. Was there a mistake in this figure? Or did the value change in the revision? Either the figure or the text needs to be corrected so they are consistent with each other.*

**Response**

 Thank you for noticing. The incorrect file was uploaded in the revised version. The file has been changed to the original, and we will be extra careful to upload the correct file during the submission process:

[Figure]

**Figure 2.** Modelled yearly Gross Primary Productivity (GPP) from 2001 to 2015 versus FLUXCOM GPP dataset (Jung et al. , 2019).

**Comment**

*Line 311: Are these differences in correlations statistically significant? The differences seem very small (0.7, 0.73, 0.74).*

**Response**

Thank you for your comment. However the UVic ESCM does not have internal variability. Any simulation with the same initial conditions, forcings, and compiler, will have identical outcomes to machine precision. This means than any change in the comparison to data is due to changes in the model structure not due to internal stochastic processes. That is, 'statistical significance' is mathematically undefined for our model structure. The UVic ESCM is purely deterministic model.

**Comment**

*Figure 4: It's Pearson's r, not Person's r. Also, the panels in this figure should be labeled with letters (a,b,c...) that can be referred to in the caption.*

**Response**

Thank you for your comment. Figure 4. Person's r has been changed to Pearson's r. New labels have been added to Figure 4 as:

[Figure]

**Figure 3.** a. FLUXCOM GPP dataset from 2000-2010, b. Seasonal GPP from 1990-2015 for Baseline, CN and CNP. c. Second line shows the global GPP from 2000-2010 for Baseline, CN and CNP. d. The third line shows the difference between Baseline, CN and CNP and FLUXCOM GPP dataset. e. Shows the correlation of Baseline, CN and CNP to FLUXCOM GPP dataset.

**Comment**

*Line 355: It's not clear what "this shift" is referring to. The previous sentence referred to a shift in broadleaf trees, not needleleaf trees.*

85

**Response**

Thank you for your comment. Line 355 has been changed to:

Needleleaf trees were reduced in North America and Europe.

**Comment**

*Line 360-365: The response to reviewers argues that it is not necessary to show a figure of PFT biomass because PFT fractions are shown, but I still think it would be helpful to show a map of biomass so this detailed description of spatial patterns could be matched with a visualization.*

**Comment**

Thank you for your comment. CNP biomass has been added to figure 6:

[Figure]

**Figure 4.** PFTs fractions in the UVic ESCM for 1980-2010, CNP minus baseline. Bottom last plot shows CNP global biomass distribution.

**Comment**

*Line 369: I would add a reference to Figure 9, which shows these patterns. Also, units of nitrogen stock should be kg N, not kg C.*

**Response**

Thank you for your comment. A reference to Figure 9 has been added. kg C has been changed to kg N.

**Comment**

*Line 374: This should also be units of N, not C*

**Response**

Thank you for your comment. C has been changed to N.

**Comment**

*Line 387-390: Reference Figure 9, which shows these patterns.*

**Response**

Thank you for your comment. A reference to Figure 9 has been added.

**Comment**

*Line 397-408: This paragraph should reference Figure 10, which shows the N2O pattern. I also found it odd that the paragraph refers to several observational datasets that are not shown in the figure but does not describe the EDGAR dataset that is shown in the figure. It would help readability if the text and the figure were more consistent with each other.*

**Comment**

115  Thank you for your comment. A reference to Figure 10 has been added. Furthermore, the following line has been added to the paragraph:

  Figure 10 shows CN and CNP $N_2O$ fluxes from 1990 to 2018. Compared to EDGAR version 6 dataset Crippa et al. (2021) our model simulates $N_2O$ fluxes relatively well, agreeing mostly in last 10 years of
120  the values. However, we observed an overestimation from 1990 to 2010.

**Comment**

 *Line 424-425: The response to reviews says that this sentence was changed to say that the underestimate was due to lack of P fertilization, but the text still says that the underestimate is due to high mineralization rate.*

125  **Response**

Thank you for your comment. You are correct. The line has been changed to:

  This underestimation is likely the result of the lack of P fertilization on land.

**Comments from reviewer 3**

**Comment**

*I'd like to thank the authors for addressing the comments and suggestions provided in the first review. Overall, your response seems reasonable and appropriate to an extent. It is good to see that you have acknowledged the reviewer's suggestion while explaining the main focus and purpose of your paper. However, we recommend that you explicitly state the same in the introduction as well.*

**Response**

Thank you for your comment. The following text has been added to the introduction:

We aim to describe a terrestrial nitrogen and phosphorus cycle adapted, developed and implemented for the UVic ESCM version 2.10. The main dynamics captured in this study are in the terrestrial system, especially vegetation. Furthermore, we intent to improve the current state of the previous N cycle implement in the UVic ESCM, develop a new P cycle and couple carbon nitrogen and phosphorus, in order to improve the carbon cycle feedbacks projections.

**Comment**

*They also provide justification for not including energy and water assessments as they were already evaluated in a recent study and adding them would not add much value to the present paper. However, it is not accurate to say "The addition of terrestrial N and P cycles had only a minor effect of these variables", as broadly discussed in Arora et al., 2020) and (Braghiere et al., 2022). I suggest briefly expanding this discussion in introduction too.*

**Response**

Thank you for your comment. The following text has been added to the last paragraph of the introduction

The addition of nutrient limitation has been observed to mainly effect the capacity of vegetation to up-take carbon (Wang et al. , 2010; Goll et al. , 2017; Wang et al. , 2020). Therefore, the accumulation of carbon in the atmosphere is enhanced, leading to increases of temperature in simulations. This temperature changes are likely to have some impact to variables sensitive to atmospheric temperature changes. Furthermore, the decrease of vegetation biomass affects variables affected by the distribution and composition of plant functional types, as changes in terrestrial albedo.

**Comment**

*The authors also mention the challenge of validating nutrient cycles in Earth system models due to the lack of observations. However, it is important to note that the author could have provided more specific details on how they validated nutrient cycles with other modeling studies and available observations to strengthen their argument. For example, in Fig 9 the authors show the IGBP global soil nitrogen right next to their map, but without any statistical evaluation. I suggest adding in the manuscript a paragraph calling for an ILAMB like tool (and reference (Collier et al., 2018)) specifically for nutrient cycles, and share about your own experience of how difficult it was to find validations tools for nutrients.*

**Response**

Thank you for your comment. The following lines has been added to section 2.5:

One of the challenges of modelling nutrients in terrestrial systems is the lack of observations and validation datasets. Furthermore, the existing range of values for N and P variables are highly uncertain. This large range in values makes it difficult to accurately tune models. Although, improvements are in sight, with new artificial intelligence derived global datasets beginning to become available (He et al. 2021). Model validation has been advancing quickly in the last decade (Spafford and MacDougall , 2021) with tools such as the International Land Model Benchmarking (Collier et al. , 2018) that significantly improves terrestrial model validation. However, there are limited variables available to compare to nutrient model development. The increase of the addition of nutrient structures in ESMs (Arora et al. , 2020) suggest the need of terrestrial nutrient validation tools to improve model accuracy in the developmental phase.

Moreover, a terrestrial nutrient model intercomparison project would unify global efforts to improve the representation of nitrogen and phosphorus in ESMs.

**Comment**

*We also appreciate your attention to detail and agree that Figure 3 should say litter N.*

**Response**

Thank you for noticing. Litter P has been changed to Litter N.

**Comment**

*Furthermore, while your biodiversity comment adds a little, we suggest that you continue to work on properly linking biodiversity with other aspects of the biogeochemical cycles in the Earth system and climate change. We recommend conducting a more thorough literature review in the introduction. Additionally, citing (Wieder et al., 2015) and (Zaehle et al., 2015) would help provide more general scientific discussions at the beginning.*

**Response**

Thank you for your comment. The following lines has been added in the introduction:

The fluxes and availability of N and P in soils depends on the interactions between soil mineral matrix, plants and microbes (Cotrufo et al. , 2013). For example, N input from atmospheric $N_2$ fixation is mediated by a specialized group of microorganisms. Furthermore, the recycling on N from plants-soil-microbes determines the availability of N for plant uptake. Overall, the land biota dynamics impacts the productivity, ecosystem resiliense and stability (Yang et al. , 2018). High diversity has been linked to enhanced vegetation productivity (Wagg et al. , 2014). The diversity in terrestrial ecosystem is determined by biological, environmental and physico-chemical processes. Anthropogenic activities and land use change emission can influence soil diversity, impacting the availability and cycling of N and P (Chen et al. , 2019). For

example, N and P fertilization, have been shown to affect soil microbial biomass and composition (Ryan et al. , 2009).

**Comment**

*We also recommend defining acronyms in their first appearance and keeping their use consistent through-out the manuscript. We noticed that the manuscript is still inconsistent with some places using nitrogen/phosphorus, while others use N/P.*

**Response**

Thank you for your comment. We have checked that acronyms are defined at their first appearance. However, if there first appearance happened to be located in the abstract, we deem necessary a second re-definition in the main text to account for readers who skip the abstract. We also checked for inconsistent use of nitrogen/phosphorus instead of N and P.

**Comment**

*Moreover, in Figure 9, we suggest that you add statistical metrics in comparison to the IGBP product and other models. At the very least, describe it in the text, as shown in Braghiere et al. (2022) Fig S13..*

**Response**

Thank you for your comment. We do no think there is a need for a statistical test for the IGBP product. We are not testing the statistical significance difference between our model outputs and the dataset in this case. The reason being that they are already clearly different in the maps, our model is below the concentration of the IGBP while our global distribution is closer to the IGBP. Hence, we only used range comparison. Regaring Fig S13. Average global distribution of leaf (a) N:P ratio, (b) C:N ratio, and (c) C:P ratio from 1994 to 2005 in ELM-FUN3.0. we do not understand how is a statistical description shown in here.

**Comment**

*On line 79, we suggest that you name and reference a few important ESMs that include N (CLM, JULES, etc.) and those that include P (ELM, CABLE). In Braghiere et al. (2022), the authors have this information right in the introduction, which you can use as a reference.*

**Comment**

Thank you for your comment. We have added the reference for the suggested models in line 76.

**Comment**

*Line 793. I suggest adding "The model does not account for uptake constrains on terrestrial vegetation, such as the Fixation and Uptake of Nutrients (FUN) model (Allen et al., 2020; Brzostek et al., 2015; Fisher et al., 2010). This includes spatial representations of mycorrhizal associations and the carbon cost of nitrogen and phosphorus uptake from soil Shi et al. , 2016; Braghiere et al. , 2021, 2022). Furtermore, we do not estimate nitrogen cost for phosphorus metabolization or viseversa.*

**Response**

Thank you for your comment. The following has been added to the discussion:

The model does not account for root uptake constrains of N and P on terrestrial vegetation. This includes spatial representations of mycorrhizal associations and the carbon cost of nitrogen and phosphorus uptake from soil (Shi et al. , 2016; Braghiere et al. , 2021, 2022).

**Comment**

*Finally, what are the next steps for the improvement of this model. Write a paragraph talking about future research.*

**Response**

245  Thank you for your comment. We do mention a list of plan improvements to the model. We clarify our
intend in line 480, it now states:

[revised manuscript text omitted]